# Optogenetics reveals Cdc42 local activation by scaffold-mediated positive feedback and Ras GTPase

Iker Lamas[iD][ᵒ], Laura Merlini[ᵒ], Aleksandar Vještica*, Vincent Vincenzetti, Sophie G. Martin[iD]*

Department of Fundamental Microbiology, Faculty of Biology and Medicine, University of Lausanne, Lausanne, Switzerland

ᵒ These authors contributed equally to this work.
* Sophie.Martin@unil.ch (SGM); Aleksandar.Vjestica@unil.ch (AV)

**Data Availability Statement:** All relevant data are within the paper and its Supporting Information files.

## Abstract

Local activity of the small GTPase Cdc42 is critical for cell polarization. Whereas scaffold-mediated positive feedback was proposed to break symmetry of budding yeast cells and produce a single zone of Cdc42 activity, the existence of similar regulation has not been probed in other organisms. Here, we address this problem using rod-shaped cells of fission yeast *Schizosaccharomyces pombe*, which exhibit zones of active Cdc42-GTP at both cell poles. We implemented the CRY2-CIB1 optogenetic system for acute light-dependent protein recruitment to the plasma membrane, which allowed to directly demonstrate positive feedback. Indeed, optogenetic recruitment of constitutively active Cdc42 leads to co-recruitment of the guanine nucleotide exchange factor (GEF) Scd1 and endogenous Cdc42, in a manner dependent on the scaffold protein Scd2. We show that Scd2 function is dispensable when the positive feedback operates through an engineered interaction between the GEF and a Cdc42 effector, the p21-activated kinase 1 (Pak1). Remarkably, this rewired positive feedback confers viability and allows cells to form 2 zones of active Cdc42 even when otherwise essential Cdc42 activators are lacking. These cells further revealed that the small GTPase Ras1 plays a role in both localizing the GEF Scd1 and promoting its activity, which potentiates the positive feedback. We conclude that scaffold-mediated positive feedback, gated by Ras activity, confers robust polarization for rod-shape formation.

## Introduction

Cell morphogenesis, proliferation, and differentiation all critically rely on polarized molecular cues. In metazoans and fungi, the Rho family guanosine triphosphatase (GTPase) Cdc42 is central for regulation of polarized cortical processes [1–3]. Cdc42 associates with the plasma membrane through a prenylated cysteine at the C-terminal CAAX motif and alternates between the active, guanosine triphosphate (GTP)-bound and the inactive, guanosine diphosphate (GDP)-bound state. As for most small GTPases, Cdc42 activation is promoted by

**Funding:** This work was supported by an ERC Consolidator grant (CellFusion) and a Swiss National Science foundation grant (310030B_176396) to SGM. AV was funded by the EMBO long-term fellowship ALTF 740-2014. The funders had no role in study design, data collection and analysis, decision to publish, or preparation of the manuscript.

**Competing interests:** The authors have declared that no competing interests exist.

**Abbreviations:** A.U., arbitrary units; B/W, black and white; Bem1, bud emergence 1; BFP, blue fluorescent protein; Cdc42, cell division cycle 42; CIB1, Cryptochrome-interacting basic helix-loop-helix 1; CIBN, N-terminal fragment of CIB1; CRIB, Cdc42- and Rac-interactive binding domain; CRY2, Cryptochrome 2; CRY2PHR, photolyase domain of Cry2; DIC, differential interference contrast; eGFP, enhanced GFP; For3, Formin 3; GAP, GTPase activating protein; GBP, GFP-binding protein; GDP, guanosine diphosphate; GEF, guanine nucleotide exchange factor; GFP, green fluorescent protein; GTP, guanosine triphosphate; GTPase, guanosine triphosphatase; iLID, improved light-induced dimer; LOV, light oxygen voltage; MAPK, mitogen activated protein kinase; PAK, p21-activated kinase; PB1, Phox and Bem1 domain; RFP, red fluorescent protein; RhoA, Ras homolog family member A; RitC, 62 C-terminal amino acids from the mammalian Rit (Ras-Like without CAAX 1 Protein Expressed In Many Tissues) protein; Rsr1, Ras-related 1; Scd, shape and conjugation deficiency; sfGFP, superfolder GFP; SH3, Src-homology 3; Tea1, Tip Elongation aberrant 1; TULIP, tunable light-inducible dimerization tag.

guanine nucleotide exchange factors (GEFs), which exchange GDP for GTP, and reversed by GTPase activating proteins (GAPs), which enhance its low intrinsic GTPase activity.

In yeast cells, in which the regulation of Cdc42 is arguably best understood, Cdc42 promotes polarized cell growth in response to internal signals during the vegetative cycle and in response to external pheromone gradients during sexual reproduction [3, 4]. Cell polarization critically relies on the local activity of Cdc42. In rod-shaped fission yeast *Schizosaccharomyces pombe* cells, Cdc42 is active at cell poles, as revealed by the Cdc42- and Rac-interactive binding domain (CRIB) probe, which specifically binds Cdc42-GTP [5]. Cdc42-GTP promotes polarized cell growth by targeting the delivery of new plasma-membrane material and cell wall–remodeling enzymes through recruitment and activation of its effectors: p21-activated kinases (PAKs), formins for nucleation of actin cables, and the exocyst complex for polarized exocytosis [4]. Cdc42 protein localizes ubiquitously at the plasma membrane in fission yeast [6–8] and only becomes enriched at cell poles as a consequence of its local activation. This local enrichment at cell poles is due to the slower lateral diffusion of Cdc42-GTP, whereas the faster-diffusing Cdc42-GDP decorates cell sides [6]. Thus, in contrast to its ubiquitous localization, Cdc42 activity is strictly polarized. The importance of local Cdc42 activity is illustrated by the observation that not only does loss of Cdc42 result in small, dense, round cells, but also its constitutive activation, or lack of inactivation, causes the formation of large, round cells [6, 9, 10]. Thus, a key question is what controls the local activation of Cdc42.

Fission yeast cells express 2 Cdc42 GEFs, which are together essential to support viability [11, 12]: Gef1, which localizes ubiquitously in the cell and in some cases accumulates at sites of growth and division, promotes cytokinesis and subsequent initiation of growth from the new cell end formed by the preceding cell division [8, 11, 13–15]; Scd1, which localizes to cell poles and division sites, is important for polarized growth during interphase [14, 16, 17]. GTP hydrolysis for Cdc42 inactivation is catalyzed by the 3 GAPs Rga3, Rga4, and Rga6. [5, 10, 18]. Whereas Rga3 colocalizes with active Cdc42 at cell tips, Rga4 and Rga6 are present at cell sides. Thus, local Cdc42 activation may in part be achieved by prelocalized GEFs and broadly distributed GAPs.

The formation of a zone of Cdc42 activity is also thought to rely on positive-feedback mechanisms that locally amplify Cdc42 activation. Positive feedbacks promote spontaneous symmetry breaking when cells establish polarity in the absence of internal or external landmarks, for instance, during spore germination or upon exposure to homogeneous pheromone signals [19–24]. Furthermore, the dynamic oscillatory patterns of Cdc42 activity during vegetative growth [13, 25] and sexual reproduction [20, 26] have uncovered the existence of negative in addition to positive-feedback regulations.

Information on the mechanisms of Cdc42 feedback regulation has been mostly gained from the budding yeast *Saccharomyces cerevisiae*, which stabilizes a single patch of Cdc42 activity prior to budding. In this organism, when the cell lacks internal positional information, a positive feedback involving the formation of a complex between Cdc42-GTP, its effector kinase (PAK), the single GEF, and a scaffold protein bridging the GEF to the PAK underlies symmetry breaking [27–29]. This complex is proposed to amplify an initial stochastic activation of Cdc42 by activating neighboring Cdc42-GDP molecules, thus promoting a winner-takes-all situation with a single cluster of Cdc42-GTP [30, 31]. Mathematical modeling supported by experimental studies has, however, indicated that modulation of specific parameters —in particular, protein exchange dynamics—can yield distinct outcomes, such as multiple zones of Cdc42 activity [6, 31, 32]. Interestingly, the positive feedback is not constitutive but can be modulated. For instance, it is intricately linked to the negative feedback, which operates through PAK-dependent phosphorylation of the GEF and acts to diminish GEF activity [33, 34]. Furthermore, an optogenetic approach to locally recruit the GEF or the scaffold showed

not only that this promotes the recruitment of endogenous copies of these molecules, indicative of positive-feedback regulation, but also that the scaffold-mediated feedback is regulated by the cell cycle, as it underlies the maintenance of a single stable site of polarity in cells about to bud, but not during early G1 phase [35]. These experiments, however, did not formally test whether Cdc42 promotes its own activation. Moreover, the possible existence of feedback regulation of Cdc42 has not been tested in other organisms.

The fission yeast is an interesting system to test whether a similar positive feedback operates. Indeed, this organism is substantially different from *S. cerevisiae*, separated by at least 300 million years of evolution [36] and with a distinct rod shape where not 1 but 2 zones of Cdc42 activity coexist. In vitro data have shown the existence of a homologous complex to that described in *S. cerevisiae*, where the scaffold protein Scd2 connects the GEF Scd1 to Pak1 [37, 38]. Although this has been widely assumed in the literature [3, 4, 6, 13, 15, 23], it remains unknown whether this complex underlies a positive feedback for the establishment of active Cdc42 zones in vivo. Notably, *scd2* deletion causes cell rounding [17, 37] but does not severely affect Cdc42 activity, in contrast to *scd1Δ* [6], implying the existence of alternative mechanisms for Cdc42 activation. Several pieces of data implicate the small GTPase Ras1 as a Cdc42 regulator upstream of Scd1 [37, 39, 40]: Ras1 binds Scd1 directly [37] and is localized to the plasma membrane and activated at the cell ends like Cdc42 [39], and its deletion causes partial cell rounding [41].

In this study, we use genetic and optogenetic approaches to dissect the modes of local Cdc42 activation. We establish the CRY2PHR-CIBN optogenetic system in fission yeast to directly demonstrate the scaffold-dependent positive feedback in Cdc42 regulation. By coupling the GEF to the PAK, we reveal that a minimal feedback system bypasses scaffold requirement and is sufficient to drive bipolar growth in cells that lack otherwise essential Cdc42 regulators. Finally, we discover that Ras1 promotes Scd1 GEF activity to modulate the feedback efficiency. Dual control by positive feedback and local Ras1 activation confers robustness to the formation of Cdc42-GTP zones at the cell poles.

## Results

### Ras1 cooperates with Scd2 for Scd1 recruitment to cell poles

Cells lacking the Cdc42 GEF Scd1 or the scaffold protein Scd2 exhibit similar, nonadditive phenotypes of widened cell shape [17]. Whereas deletion of both Scd1 and Gef1 is lethal [11], we noticed that cells lacking *scd2* and *gef1* are viable (S1A Fig), which suggested that other mechanisms for Scd1 recruitment and/or activation exist. Several pieces of data implicated the small GTPase Ras1 as positive regulator of Cdc42 acting upstream of Scd1 [37, 40, 42]. Indeed, we find that *ras1* is required for viability of *scd2Δ gef1Δ* double-mutant cells (Table 1 and S1 Table) but not *gef1Δ* or *scd2Δ* single mutants (S1B and S1C Fig). We conclude that Cdc42 activation by Scd1 relies on both Scd2 and Ras1. In agreement with this view, Scd1-3GFP cortical localization was strongly reduced in *scd2Δ* and *ras1Δ* single mutants but undetectable in *scd2Δ ras1Δ* double-mutant cells (Fig 1A and 1B). *scd2Δ ras1Δ* cells also exhibited strongly reduced levels of Cdc42-GTP, as detected by CRIB-GFP, in line with the reported *scd1Δ* phenotype [6] (Fig 1C and 1D). Although these cells were nearly round (Fig 1E), they remained viable because of Cdc42 activation by Gef1, which colocalized with Cdc42-GTP to short-lived, dynamic patches formed at the membrane (Fig 1F and 1G, S1 Movie). This indicates that Gef1 is sufficient for Cdc42 activation to support viability, but not for the stabilization of zones of active Cdc42. Our results indicate that Scd2 and Ras1 cooperate to promote Scd1 recruitment and stable Cdc42 activity zones at cell poles.

**Table 1. Genetic interaction of *scd1Δ*, *scd2Δ*, *ras1Δ*, and *gef1Δ*.** Viability or lethality of indicated genotypes was assessed by tetrad dissection of crosses described in S1 Table.

| Mutant | Viability |
|---|---|
| *scd1Δ gef1Δ* | Lethal [11] |
| *scd1Δ gef1-3GFP pak1-GBP* | Viable |
| *scd2Δ gef1Δ* | Viable |
| *scd2Δ ras1Δ* | Viable |
| *ras1Δ gef1Δ* | Viable |
| *scd2Δ gef1Δ ras1Δ* | Lethal |
| *scd2$^{K463A}$ gef1Δ ras1Δ* | Lethal |
| *scd2Δ gef1Δ ras1Δ pak1-GBP scd1-3GFP* | Viable |
| *scd2Δ gef1Δ ras1Δ pak1-GBP scd1-1xGFP* | Viable |
| *scd2$^{K463A}$ gef1Δ ras1Δ pak1-GBP scd1-3GFP* | Viable |
| *scd2Δ gef1Δ ras1Δ tea1-GBP scd1-3GFP* | Lethal |
| *scd2Δ gef1Δ ras1Δ scd1-GBP for3-3GFP* | Lethal |
| *scd2Δ gef1Δ ras1Δ pak1$^{N-term}$-GBP scd1-3GFP* | Viable |
| *scd2Δ gef1Δ ras1Δ pak1$^{KRKR}$-GBP scd1-3GFP* | Viable |

## Acute recruitment of proteins to the cell cortex by optogenetics

To probe whether Cdc42 activity is regulated through positive feedback, we adapted the CRY2-CIB1 optogenetic system to acutely recruit active Cdc42 to the plasma membrane in response to light [43]. CRY2 is a blue light–absorbing photosensor that, in the photoexcited state, binds the CIB1 partner protein [44]. We used the minimal interacting domains of CRY2 (CRY2PHR, aa1-498) and CIB1 (CIBN, aa1-170) as the core components of our optogenetic system (Fig 2A) [43]. We tagged each moiety with a distinct fluorophore and targeted CIBN to the plasma membrane using an amphipathic helix from the mammalian protein Rit (RitC, Fig 2A and 2B). Hereafter, this basic setup is referred to as the Opto system. We first characterized the activation requirements and dynamics of recruitment of CRY2PHR to CIBN-RitC. In absence of blue-light stimulation (dark), CRY2PHR-mCherry localized to the cytosol and nucleus (Fig 2B). Upon blue-light ($\lambda$ = 488 nm) stimulation, cytosolic CRY2PHR rapidly relocalized to the plasma membrane (Fig 2B). Cortical recruitment of CRY2PHR was strictly dependent on CIBN and specific to blue light (Fig 2B, S2A Fig), indicating that the heterologous CRY2PHR does not interact with fission yeast cortical proteins. Relocalization of CRY2PHR occurred rapidly (Fig 2C and 2D), with kinetics largely independent from the duration of blue-light stimulation: photoactivation by 30 pulses of 50 ms, 22 pulses of 250 ms, or 17 pulses of 500 ms over 15 s all resulted in indistinguishable recruitment half-times of 0.85 ± 0.25 s, 0.82 ± 0.23 s, and 0.85 ± 0.21 s, respectively (Fig 2E and 2F). Since CRY2PHR activation dynamics were constant under varying blue-light regimes, we conclude that monitoring the localization of GFP-tagged endogenous proteins that require distinct blue-light exposure times will nevertheless lead to identical activation of the optogenetic module. Thus, the Opto system works efficiently for acute and rapid plasma-membrane recruitment in fission yeast cells.

To regulate Cdc42 activity at the plasma membrane, we fused the CRY2PHR-mCherry with a constitutively GTP-bound Cdc42$^{Q61L}$ allele, which also lacked the C-terminal CAAX box (Fig 2A). This cytosolic Cdc42$^{Q61L, \Delta CAAX}$ allele was largely nonfunctional, as cells retained their rod shape and only slightly decreased their aspect ratio (S2B Fig). We note, however, that these cells were largely monopolar, suggesting a mild dominant-negative effect (S2B Fig). We combined this construct with CIBN-RitC to build the optogenetic module we refer to as Opto$^{Q61L}$. Opto$^{Q61L}$ behaved similarly to the Opto system: First, the amplitude of the cortical

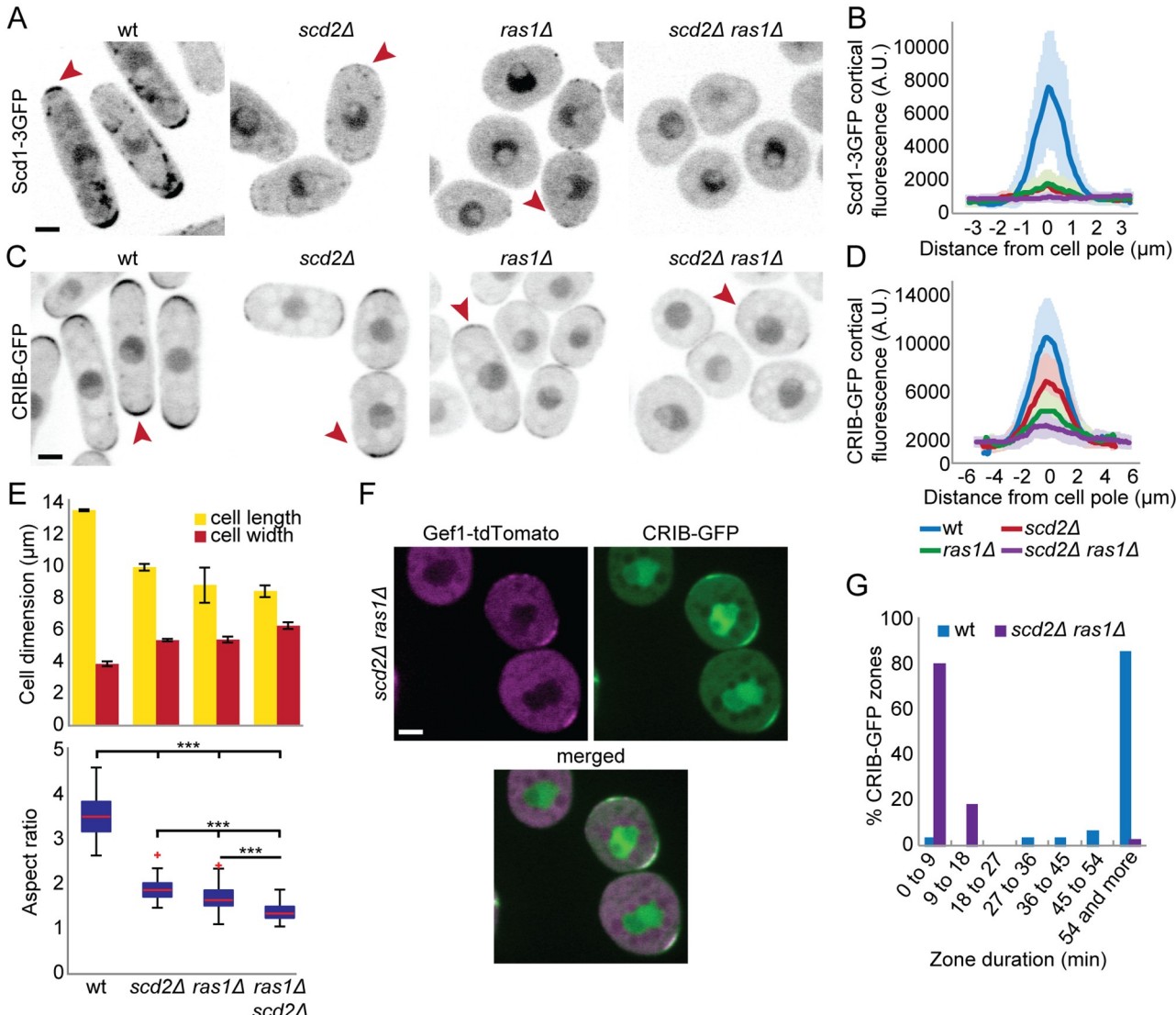

**Fig 1. Ras1 and Scd2 cooperate for Scd1 recruitment to cell poles.** (A-D) Localization of Scd1-3GFP (A-B) and CRIB-GFP (C-D) in wt, *ras1Δ*, *scd2Δ*, and *ras1Δ, scd2Δ* cells. (A) and (C) show representative B/W inverted images; (B) and (D) show cortical tip profiles of Scd1-3GFP and CRIB-GFP fluorescence; *n* = 24 and 26 cells, respectively. Thick line = average; shaded area = standard deviation. (E) Mean cell length and width at division (top) and aspect ratio (bottom) of strains as in (C) (*n* > 30 for 3 independent experiments). Bar graph error bars show standard deviation; box plots show the ratio between cell length and cell width. On each box, the central mark indicates the median; the bottom and the top edges indicate the 25th and 75th percentiles, respectively; the whiskers extend to the most extreme data points not considering outliers, which are plotted individually using the red "+" symbol. *** indicates $2.8e^{-90} \leq p \leq 1.9e^{-10}$. (F) Colocalization of CRIB-GFP and Gef1-tdTomato in *ras1Δ scd2Δ* double-mutant cells. Bar = 2 μm. (G) Lifetime of CRIB-GFP cortical zones over 1 h, in wt and *ras1Δ scd2Δ* cells (*n* = 32 and 39 zones respectively). All underlying numerical values are available in S1 Data. A.U., arbitrary units; B/W, black and white; wt, wild type.

recruitment was similar for both systems, indicating that Cdc42$^{Q61L}$ does not impair the recruitment of CRY2PHR to the cortex (Fig 2E). Second, recruitment of Opto$^{Q61L}$ to the plasma membrane was also very rapid, with half-times only marginally higher than those of the Opto system (0.98 ± 0.3 s; 0.91 ± 0.27 s; 0.96 ± 0.3 s for the 3 activation cycles defined above, respectively) (Fig 2F). The Opto$^{WT}$ system, in which wild-type (wt) Cdc42 lacking its CAAX box was linked to CRY2PHR-mCherry, behaved similarly. The nonnative recruitment of active Cdc42 to the plasma membrane by the Opto$^{Q61L}$ system raises the question of whether it will be able to activate effectors at the cortex. We had shown in earlier work that

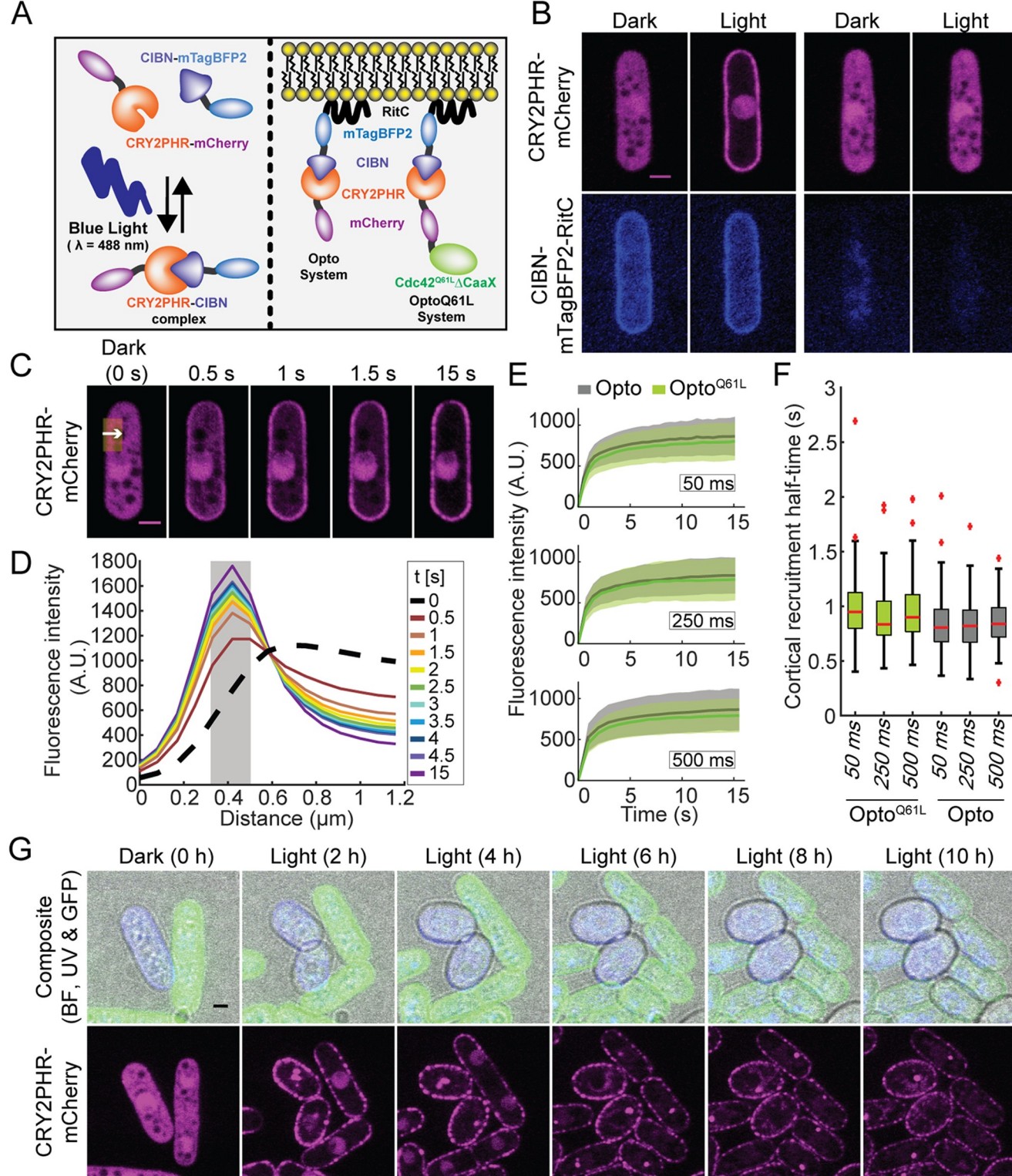

**Fig 2. Acute cortical recruitment of protein by optogenetics.** (A) Principle of the blue light–dependent CRY2PHR-CIBN complex formation (left) and configuration of heterologous synthetic proteins implemented in *S. pombe* (right). (B) Blue light–and CIBN-dependent cortical recruitment of CRY2PHR-mCherry. Note that mixtures of sample and control cells were used for this and all optogenetic experiments (see S3 Fig and Methods). The left panels center on a cell expressing both CIBN and CRY2PHR. The right panels center on a cell expressing only CRY2PHR. (C) Plasma-membrane recruitment of the Opto system in response to periodic 50-ms blue-light (λ = 488 nm, 30 cycles) stimulation. The white arrow within the yellow ROI (≈1.25 μm by 3 μm) indicates the region quantified in (D). (D) Profiles extracted from the ROI highlighted in (C); the gray area indicates the plasma-membrane position, defined as the 3 pixels surrounding the peak fluorescence intensity at the end of the time lapse. These pixel values were averaged and displayed over time to display the plasma-membrane recruitment dynamics shown in (E). (E) Plasma-membrane recruitment dynamics (extracted from the signal in the gray area in (C)) of Opto (gray) and Opto$^{Q61L}$ (green) in response to periodic 50 ms (top), 250 ms (middle), and 500 ms (bottom) blue light (λ = 488 nm) pulses (*N* = 3, *n* = 30

cells per experiment). Thick line = average; shaded area = standard deviation. (F) Plasma-membrane recruitment half-times for the Opto and Opto$^{Q61L}$ systems. On each box, the central mark indicates the median; the bottom and the top edges indicate the 25th and 75th percentiles, respectively; the whiskers extend to the most extreme data points not considering outliers, which are plotted individually using the red "+" symbol. (G) Blue light–dependent induction of isotropic growth in Opto$^{Q61L}$ (blue), but not Opto$^{WT}$ (green) cells photoactivated at 10-min interval (GFP, RFP, and BF channels were acquired every 10 min; UV channel every 1 h). Note that the patchy appearance of CRY2PHR-mCherry is likely due to the clustering properties of this protein [45]. Bars = 2 μm. All underlying numerical values are available in S2 Data. A.U., arbitrary units; BF, bright-field; ROI, region of interest.

Cdc42 alleles binding the plasma membrane through nonnative domains, including the RitC amphipathic helix, are sufficiently functional for polarity establishment [6], suggesting that Opto$^{Q61L}$ may provide sufficient Cdc42 activity at the cell cortex. As a proof of principle, we mixed Opto$^{Q61L}$ and Opto$^{WT}$ cells and performed long-term imaging: Opto$^{Q61L}$ cells became round within 6 h of periodic blue-light stimulation, whereas Opto$^{WT}$ cells continued growing in a polarized manner (Fig 2G, S2 Movie). This transition from rod to round shape is evidence of isotropic growth triggered by the recruitment of Cdc42 activity to the plasma membrane in a light-dependent manner. We thus further used Opto$^{Q61L}$ as a tool to dissect the proximal events after Cdc42 activation.

## Cdc42-GTP promotes a positive feedback that recruits its own GEF

We investigated the localization of Cdc42-GTP regulators and effectors in Opto$^{Q61L}$ cells. The Cdc42-GTP sensor CRIB, the PAK-family kinase Pak1, the scaffold protein Scd2, and the GEF Scd1, each tagged with GFP, localize to the poles of interphase wt cells. In cells where Opto$^{Q61L}$ was kept inactive, because of either absence of CIBN-RitC or dark conditions, Scd1-3GFP, Scd2-GFP, Pak1–superfolder GFP (sfGFP), and CRIB-3GFP were all observed at cell tips with some-what reduced intensities, as compared to wt cells (t0 of Fig 3A–3C). Expression of CRY2PHR-Cdc42$^{Q61L}$ resulted in partial sequestering of Scd2 to the nucleus, which may explain the decreased cortical levels of Scd1 and Pak1, as well as the slight decrease in aspect ratio and mono-polar growth of the Opto$^{Q61L}$ cells (S2B Fig). This observation suggests that Scd2 interacts with GTP-locked Cdc42 even when not at the membrane. Importantly, all components of the Cdc42 module were excluded from the lateral cell cortex in cells with the inactive Opto$^{Q61L}$ system.

To probe whether the isotropic growth induced by Opto$^{Q61L}$ correlated with local recruit-ment of polarity factors, we first imaged the scaffold Scd2-GFP, which directly binds Cdc42-GTP [38, 46], during long time-lapse acquisitions. Scd2 was recruited to cell sides by Opto$^{Q61L}$ and diminished at the cell tips (Fig 3A). Note that CRY2PHR displays a patchy appearance at the cortex, which is particularly prominent in long time-lapse acquisitions. This is likely due to its clustering properties, as previously described [45, 47, 48]. We exploited CRY2PHR clustering to show that Scd2-GFP distribution was spatially well correlated with clusters of Opto$^{Q61L}$ at cell sides (Fig 3A and 3B), in agreement with the notion that it is directly recruited by Opto$^{Q61L}$.

To probe the immediate effect of Cdc42 activity cortical relocalization, we focused on the response of cells in the first seconds and minutes after light-induced recruitment of Opto$^{Q61L}$ to the cell cortex. In all subsequent optogenetic experiments, sample cells expressing both Opto$^{Q61L}$ and a GFP-tagged protein of interest were mixed with control cells containing only the GFP tag or only cytosolic CRY2PHR-mCherry, which were used as negative control and for correction and normalization of the measured fluorescence signals at the lateral cell cortex (S3 Fig; see Methods). Activation of Opto$^{Q61L}$ by blue light promoted a rapid relocalization to cell sides—and concomitant depletion from cell tips—of CRIB, Pak1, and Scd2, each of which binds Cdc42-GTP directly (Fig 3C; S4A Fig for individual traces; S3 Movie) [37, 38, 46, 49]. In agreement with these proteins binding the constitutively active Cdc42$^{Q61L}$ directly, Scd2 recruitment to Opto$^{Q61L}$ required neither Scd1 nor the other GEF Gef1 (S5A and S5B Fig). Pak1 was also recruited to Opto$^{Q61L}$ independently of Scd1 (S5C and S5D Fig). The scaffold

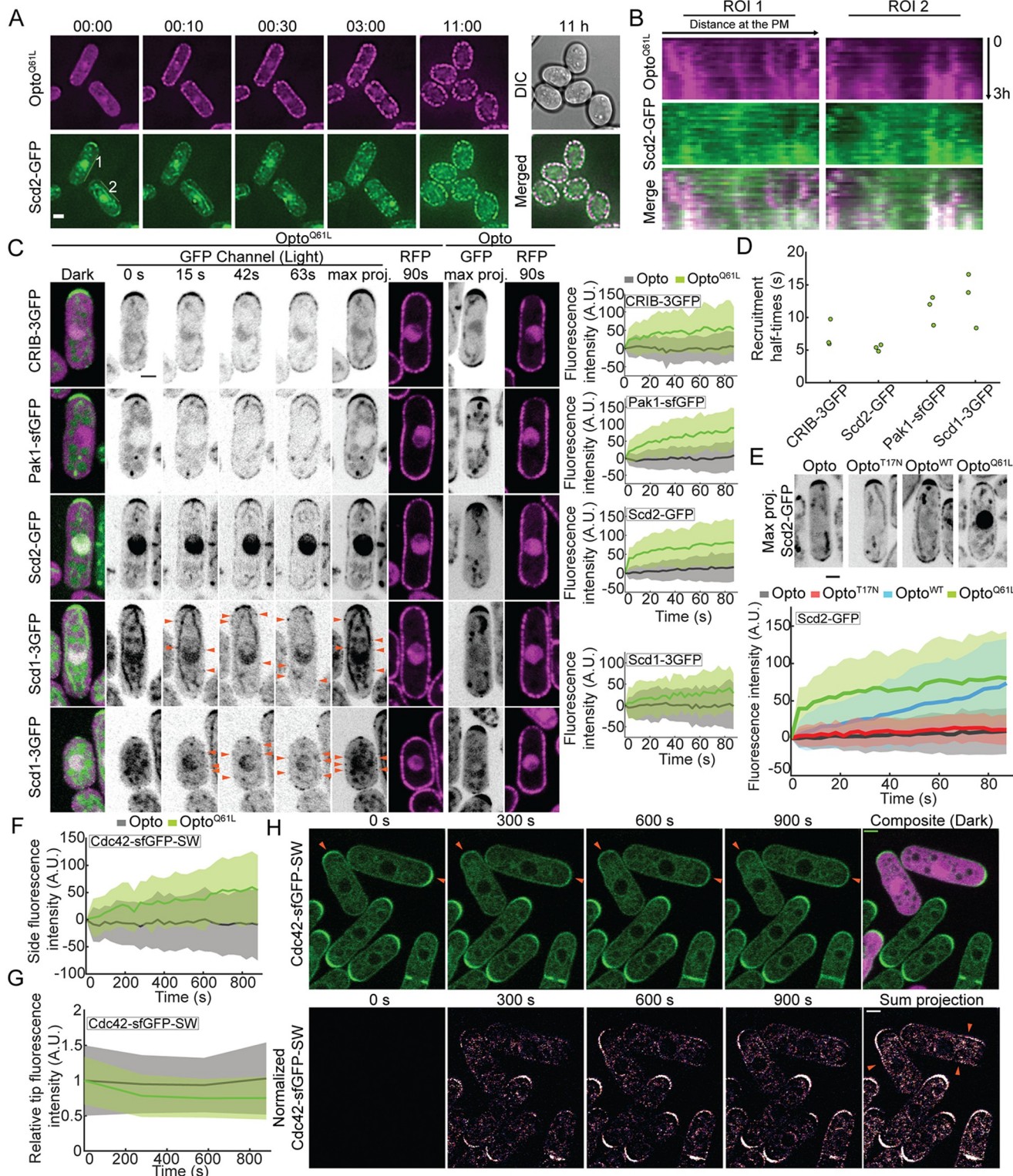

**Fig 3. Visualizing the positive feedback triggered by Cdc42-GTP.** (A) Blue light–dependent Scd2-GFP relocalization to Opto$^{Q61L}$ foci during the transition from polar to isotropic growth (GFP, RFP, and DIC channels were acquired every 10 min). ROIs (2 pixel width) highlight cortical regions from where kymographs shown in (B) were generated. Time is shown in hh:mm. (B) Kymographs generated from ROIs shown in (A) over the first 3 h of time-lapse acquisition. x-axes of kymographs represent distance at the cell side and y-axes represent time. (C) Opto$^{Q61L}$-induced cell-side relocalization of CRIB-3GFP probe and endogenously tagged Pak1-sfGFP, Scd2-GFP, and Scd1-3GFP in otherwise wt cells (B/W inverted images). Merged color images on the left show the dark, inactive state of Opto$^{Q61L}$ cells. The GFP max projection ("max proj.") images show GFP maximum-intensity projections of 30 time points over 90 s and illustrate best the side recruitment of GFP-tagged probes. Orange arrowheads point to lateral Scd1-3GFP signal. RFP images show the cortical recruitment of CRY2PHR-mCherry-Cdc42$^{Q61L}$ (Opto$^{Q61L}$) and CRY2PHR-mCherry (Opto) at the end of the time lapse. Quantification of GFP signal intensity at cell sides is shown on the right. $N = 3$; $n > 20$ cells per experiment; $p^{CRIB-3GFP} = 2.9e^{-13}$; $p^{Pak1-sfGFP} = 4.8e^{-22}$; $p^{Scd2-GFP} = 3.1e^{-18}$; $p^{Scd1-3GFP}$

= 1.4e$^{-04}$. For Scd1-3GFP (bottom 2 rows), 2 different examples of normal-sized (top) and small cells (bottom) are shown. The internal globular and filamentous signal is due to cellular autofluorescence (see S3B Fig for examples of unlabeled cells imaged in the same conditions). (D) Cell-side relocalization half-times of CRIB-3GFP, Scd2-GFP, Scd1-3GFP, and Pak1-sfGFP. Average half-times derived from 3 independent experimental replicates are plotted. (E) Relocalization to cell sides requires Cdc42 activity. (Top) Max projection images of Scd2-GFP (B/W inverted images) over 90-s illumination in Opto, Opto$^{T17N}$, Opto$^{WT}$, and Opto$^{Q61L}$ cells. (Bottom) Scd2-GFP relocalization dynamics at the sides of Opto, Opto$^{T17N}$, Opto$^{WT}$, and Opto$^{Q61L}$ cells. $N = 3$ experiments; $n > 20$ cells per experiment, except for the Opto sample, in which $N = 6$ independent experiments ($1 \times N = 3$ in parallel to Opto$^{Q61L}$ and Opto$^{WT}$, and $1 \times N = 3$ for Opto$^{T17N}$). The Opto$^{Q61L}$ trace is the same as in (C). (F) Opto$^{Q61L}$-induced cell-side accumulation of endogenous Cdc42-sfGFP$^{SW}$ in otherwise wt cells. $N = 3$; $n > 20$ cells per experiment; $p = 4.2e^{-09}$. (G) Opto$^{Q61L}$-induced decrease in Cdc42-sfGFP$^{SW}$ tip signal over time ($p^{OptoQ61LVsOpto} = 0.037$). Note that measurements were performed at every 5-min time point only. (H) Opto$^{Q61L}$-induced relocalization of endogenous Cdc42-sfGFP$^{SW}$ in otherwise wt cells. (Top) Mixtures of Opto$^{Q61L}$ (purple) and GFP control cell expressing Cdc42-sfGFP$^{SW}$. Note tip accumulation Cdc42-sfGFP$^{SW}$ at t0, which is lost over time in Opto$^{Q61L}$ cells (arrowheads). (Bottom) Time lapse as in top row, but normalized by subtraction of initial time-point signal and pseudocolored. Note side signal in Opto$^{Q61L}$ cells (arrowheads), which represents gain of fluorescence intensity in respect to the initial time point (see Methods). The strong signal at the tips of wt cells is due to cell growth. In all graphs, thick line = average; shaded area = standard deviation. Bars = 2 μm. All underlying numerical values are available in S3 Data. A.U., arbitrary units; B/W, black and white; DIC, differential interference contrast; PM, plasma membrane; ROI, region of interest; wt, wild type.

protein Scd2-GFP relocalized with the fastest dynamics ($t_{1/2}$ Scd2 = 5.4 ± 0.5 s; Fig 3D) along with CRIB ($t_{1/2}$ CRIB = 7.3 ± 2.1 s), consistent with Scd2 binding Cdc42-GTP already in the cytosol. We proceeded to test whether the observed Scd2-GFP dynamics are indeed caused by acute increase of active Cdc42 at the plasma membrane. We repeated the Scd2-GFP recruitment experiments using either Opto$^{WT}$ (described above) or Opto$^{T17N}$, in which a constitutively GDP-bound Cdc42$^{T17N, \Delta CAAX}$ allele is recruited to the cortex by light. Opto$^{T17N}$ did not cause notable relocalization of Scd2 (Fig 3E, S4B Fig), indicating that Cdc42 activity is essential for Scd2 recruitment. Unexpectedly, although Opto$^{WT}$ also did not have any overt effect on cell shape (see Fig 2G), it induced a noticeable recruitment of Scd2 to the cell cortex, which was, however, slower and less marked than in Opto$^{Q61L}$ cells (Fig 3E, S4B Fig), indicating some activation of Opto$^{WT}$ at the cell cortex. We hypothesize that the clustering of Cdc42 by CRY2 is sufficient to weakly activate it, as has been previously observed for RhoA [45]. In summary, ectopic optogenetic recruitment of Cdc42 activity to the cell cortex leads to rapid recruitment of the scaffold Scd2 and effector proteins.

The GEF Scd1 relocalized to cell sides with dynamics similar to Pak1 ($t_{1/2}$ Scd1 = 13.0 ± 4.2 s; $t_{1/2}$ Pak1 = 11.3 ± 2.2 s; Fig 3C and 3D, S4A Fig), and its levels at cell tips decreased (see Fig 4B). The observation that active Cdc42 recruits its own activator Scd1 along with the scaffold protein Scd2 and the effector protein Pak1 is in agreement with the existence of a positive feedback regulating Cdc42 activity in vivo. Furthermore, the endogenous Cdc42-sfGFP$^{SW}$ protein, which is functional and normally enriched at cell poles [6], accumulated at the lateral cortex of Opto$^{Q61L}$ cells upon blue-light illumination, with slower kinetics (Fig 3F and 3H, S4C Fig). The increase of endogenous Cdc42 protein at the cell sides was mirrored by a gradual loss of its enrichment at the cell tips (Fig 3G and 3H). The loss from cell tips and accumulation at cell sides are best appreciated from normalized images, in which the initial Cdc42 fluorescence signal is subtracted from post-illumination time points (Fig 3H, bottom). We conclude that Cdc42$^{Q61L}$ at cell sides triggers a positive feedback and competes against the native polarity sites at the cell tips for Scd1, Scd2, and Pak1 polarity factors as well as endogenous Cdc42. Consistent with the localization of Scd1, Scd2, and Pak1 to zones of active Cdc42 at cell tips, we infer that wt Cdc42-GTP trigger such positive feedback in interphase cells.

## The Scd2 scaffold is essential for Cdc42-GTP feedback regulation

Scd2 is rapidly recruited by Cdc42-GTP (Fig 3A–3E); forms complexes in vitro with Cdc42, Pak1, and Scd1 [38]; and promotes Scd1 recruitment to the cell tip (Fig 1). These 3 observations prompted us to test the role of Scd2 as a scaffold for the Cdc42 positive feedback. In scd2Δ cells, Opto$^{Q61L}$ induced relocalization of CRIB and Pak1, which bind Cdc42-GTP directly (Fig 4A, S6A Fig, S4 Movie). However, we did not detect any cell-side accumulation

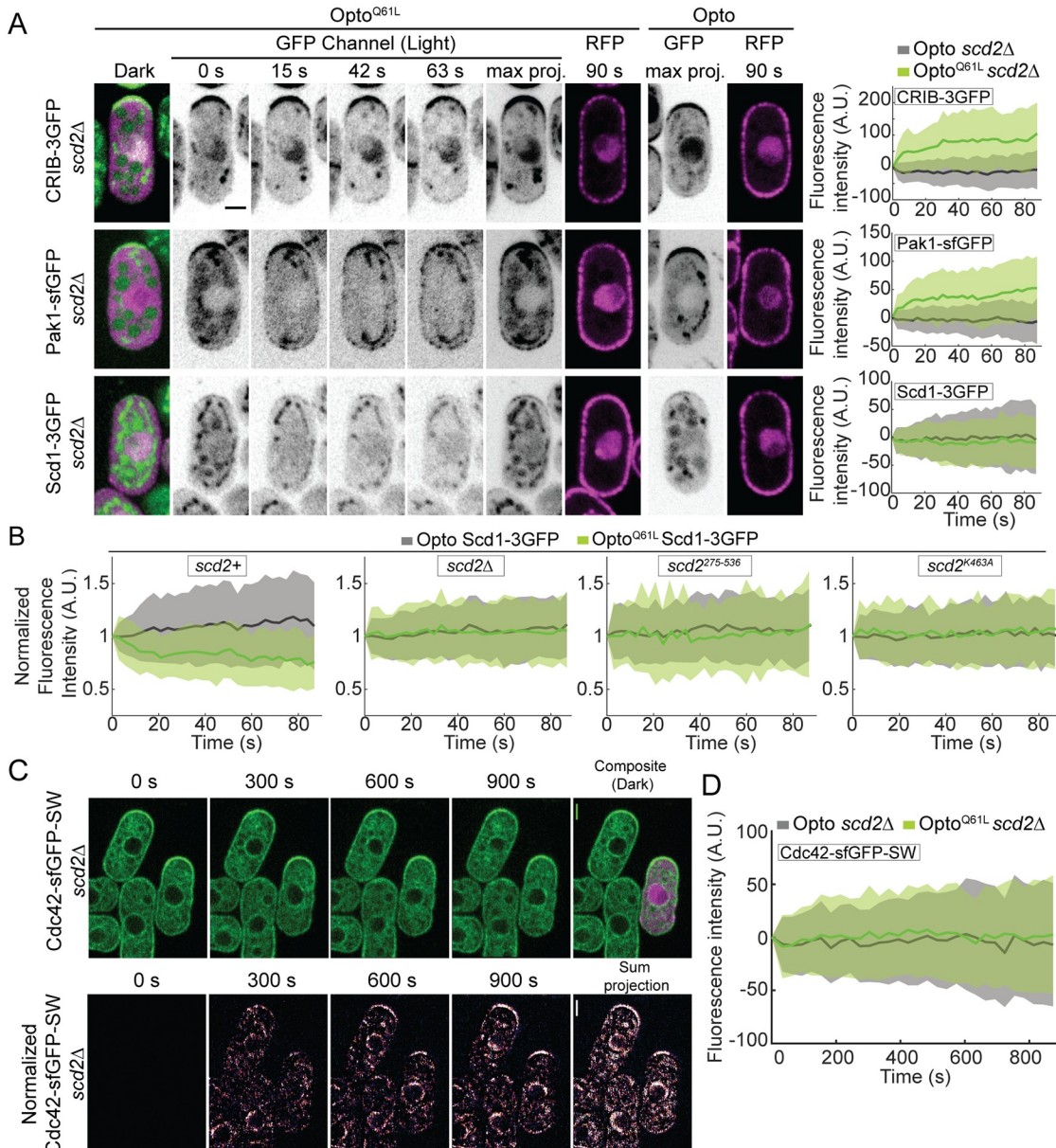

**Fig 4. Scd2 is essential for the Cdc42-GTP-triggered positive feedback.** (A) In *scd2Δ* cells, Opto<sup>Q61L</sup> induces CRIB and Pak1 recruitment but fails to recruit its GEF Scd1. Data layout as in Fig 3C. $N = 3$; $n > 20$ cells per experiment; $p^{CRIB-3GFP} = 1.4e^{-18}$; $p^{Pak1-sfGFP} = 8.4e^{-15}$; $p^{Scd1-3GFP} = 0.2$. (B) Scd1-3GFP signal at cell tips over time in Opto<sup>Q61L</sup> and Opto *scd2+* ($p = 1.6e^{-09}$), *scd2Δ* ($p = 0.5$), *scd2^{275-536}* ($p = 0.1$), and *scd2^{K463A}* cells ($p = 0.8$). $N = 3$; $n > 15$ cells. (C) Mixtures of *scd2Δ* Opto<sup>Q61L</sup> (purple) and *scd2Δ* control cell expressing Cdc42-sfGFP<sup>SW</sup>. Data presented as in Fig 3H. Note that there is little change in Cdc42-sfGFP<sup>SW</sup> distribution in *scd2Δ* Opto<sup>Q61L</sup> cells. (D) Opto<sup>Q61L</sup> fails to induce cell-side accumulation of endogenous Cdc42-sfGFP<sup>SW</sup> in *scd2Δ* cells. $N = 3$; $n > 20$ cells per experiment; $p = 0.32$. Bars = 2 μm. All underlying numerical values are available in S4 Data. A.U., arbitrary units; max proj., maximum projection.

of Scd1 (Fig 4A, S6A Fig, S4 Movie). Because the Scd1 cortical signal is weak in *scd2Δ* cells, we were concerned that we may fail to detect low levels of Scd1 at cell sides. We thus also assessed whether Scd1 from cell tips was depleted by Opto<sup>Q61L</sup> side recruitment, which would indicate competition (Fig 4B). In contrast to the situation in *scd2+* cells, upon Opto<sup>Q61L</sup> activation in *scd2Δ* mutants, Scd1-3GFP tip levels remained constant, supporting the notion that Scd1 does not relocalize in these cells. Similarly, endogenous Cdc42-sfGFP<sup>SW</sup> protein did not change

localization upon blue-light illumination in Opto$^{Q61L}$ cells lacking *scd2* (Fig 4C and 4D, S6B Fig), in contrast to *scd2*$^+$ cells. We conclude that the scaffold Scd2 is required for Scd1 GEF recruitment to Cdc42-GTP, which underlies a positive feedback on Cdc42.

To further probe the importance of Scd2-Cdc42 and Scd2-Scd1 interactions in the positive feedback, we constructed 3 *scd2* mutant alleles at the endogenous genomic locus (Fig 5A): (1) an N-terminal fragment (aa1-266) sufficient to bind Cdc42-GTP, *scd2*$^{1-266}$; (2) a truncation lacking this region, *scd2*$^{275-536}$ [46]; and (3) the point mutant *scd2*$^{K463A}$, predicted to impair Scd1 binding [50]. All 3 alleles were expressed at slightly higher levels than wt, exhibited loss of function, and produced cells with dimensions similar to *scd2Δ*, but each mutant showed a distinct localization (Fig 5B and 5C). Scd2$^{K463A}$ localized correctly to cell tips; Scd2$^{1-266}$ showed strong, but irregularly positioned, cortical signal; Scd2$^{275-536}$ was cytosolic. Both Scd2$^{K463A}$ and Scd2$^{1-266}$ were recruited to cell sides by Opto$^{Q61L}$ with kinetics similar to wt Scd2 (Fig 5D, S7A Fig). Conversely, in Opto$^{Q61L}$ cells, Scd2$^{275-536}$ did not accumulate in the nucleus or at the cell sides in response to light (Fig 5D, S7A Fig). These data indicate that Cdc42-GTP recruits Scd2, either directly or with the help of Pak1, which also binds Scd2 N terminus. The more robust localization of Scd2$^{K463A}$ versus Scd2$^{1-266}$ may be due to presence of the PX domain (Fig 5A), which has been shown to interact with phosphoinositides in budding yeast [51, 52].

As expected, the cytosolic Scd2$^{275-536}$ fragment failed to induce Scd1 relocalization to cell sides in Opto$^{Q61L}$ cells. Similarly, Scd2$^{K463A}$, which is itself localized correctly, was unable to induce cell-side relocalization of Scd1 (Fig 5E, S7B Fig) or loss of Scd1 tip signal (see Fig 4B). In agreement with these results, Scd1 levels at cell poles were strongly reduced in *scd2*$^{K463A}$ mutant cells and undetectable when *ras1* was also deleted (Fig 5F). *scd2*$^{K463A}$ was also synthetic lethal in combination with *gef1Δ* and *ras1Δ* (Table 1 and S1 Table). Together, these results show that the recruitment of Scd1 to constitutively activated Cdc42 at nonnative sites fully depends on Scd2, which itself depends on Cdc42-GTP. Thus, Scd2 is the scaffold that promotes positive feedback by linking Cdc42-GTP to its GEF Scd1.

## An artificial Scd1-Pak1 bridge is sufficient to sustain bipolar growth

To test the importance of the positive feedback for cell polarity in vivo, we engineered a simplified system to bypass the Scd2 function of linking the GEF Scd1 to PAK. We used the nanomolar affinity between GFP and GFP-binding protein (GBP; [53]) to create an artificial Scd1-Pak1 bridge. We tagged Scd1 and Pak1 at their endogenous genomic loci with 3GFP and GBP-mCherry, respectively, and generated strains coexpressing these alleles. The forced interaction of Scd1 with Pak1 (henceforth called Scd1-Pak1 bridge) in otherwise wt cells; *scd2Δ*, *gef1Δ*, *ras1Δ* single mutants; and *scd2Δ gef1Δ*, *gef1Δ ras1Δ*, or *scd2Δ ras1Δ* double mutants did not affect the ability of cells to form colonies at different temperatures (S8A Fig) and was sufficient to restore a near-normal cell shape to *scd2Δ* and *scd2*$^{K463A}$ cells (Fig 6A), suggesting that the main function of Scd2 is to mediate Scd1-PAK complex formation.

Remarkably, the Scd1-Pak1 bridge suppressed the lethality of *scd2Δ gef1Δ ras1Δ* and *scd2*$^{K463A}$ *gef1Δ ras1Δ* triple-mutant cells (Fig 6B, S8A Fig; Table 1 and S1 Table). Lethality suppression occurred specifically by linking Scd1 to the PAK, but not to another cell tip–localized protein such as Tea1 or another Cdc42 effector such as For3 (Table 1 and S1 Table) [54, 55]. This suggests that rescue requires proximity of the GEF to the PAK, which cannot be substituted by another Cdc42 effector. Strikingly, *scd2Δ gef1Δ ras1Δ* triple-mutant cells expressing the Scd1-Pak1 bridge were not only viable but also formed rod-shaped cells and grew in a bipolar manner (Fig 6C, S5 Movie). These cells were significantly shorter and only slightly wider than wt cells, with an aspect ratio similar to that of *ras1Δ* and *scd2Δ ras1Δ* mutants expressing the bridge (Fig 6D, S8B Fig), suggesting a function of Ras1 not rescued by the

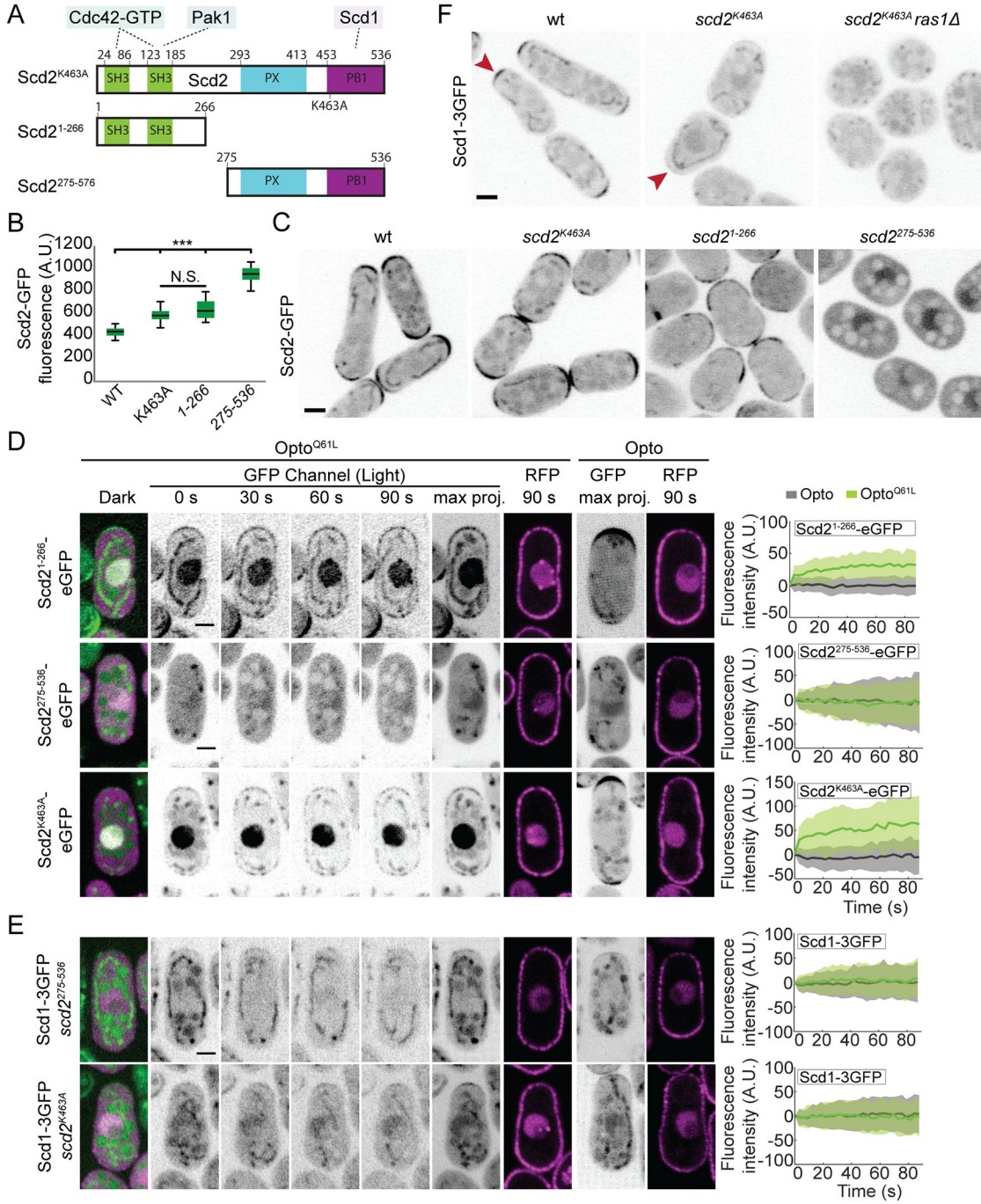

**Fig 5. Structure-function dissection of Scd2 in positive feedback.** (A) Schematics of Scd2 showing interaction domains with Cdc42-GTP (SH3 1 and 2), Pak1 (SH3 2), and Scd1 (PB1) [37, 38, 46], with point mutation used to block Scd1 binding (top; $Scd2^{K463A}$; [50]) and N- and C-terminal $Scd2^{1-266}$ and $Scd^{275-536}$ fragments (bottom). The PX domain likely binds phosphoinositides [51, 52]. (B) Quantification of Scd2-GFP total fluorescence in strains as in (C). (C) Localization of Scd2-GFP (B/W inverted images) in cells expressing different alleles integrated at the endogenous *scd2* locus: $scd2^{wt}$, $scd2^{K463A}$, $scd2^{1-266}$, and $scd2^{275-536}$. (D) Opto$^{Q61L}$ induces the relocalization of Scd2$^{1-266}$ and Scd2$^{K463A}$ but fails to recruit Scd2$^{275-536}$. Data layout as in Fig 3C. $N = 3$; $n > 20$ cells; $p^{scd2(1-266)\text{-eGFP}} = 3.9e^{-23}$, $p^{scd2(275-536)\text{-eGFP}} = 0.69$ and $p^{Scd2K463A\text{-eGFP}} = 4.4e^{-21}$. (E) Opto$^{Q61L}$ fails to recruit its own GEF Scd1 in $scd2^{275-536}$ and $scd2^{K463A}$ mutants. Data layout as in Fig 3A. $N = 3$; $n > 20$ cells; in $scd2^{275-536}$, $p^{Scd1\text{-3GFP}} = 0.2$; in $scd2^{K463A}$, $p^{Scd1\text{-3GFP}} = 0.3$. (F) Localization of Scd1-3GFP (B/W inverted images) in wt, $scd2^{K463A}$, and $scd2^{K463A}$ $ras1\Delta$ cells. In all graphs, thick line = average; shaded area = standard deviation. Bars = 2 μm. All underlying numerical values are available in S5 Data. A.U., arbitrary units; B/W, black and white; max proj., maximum projection; N.S., not significant; PB1, Phox and Bem1 domain; SH3, Src-homology 3; wt, wild type.

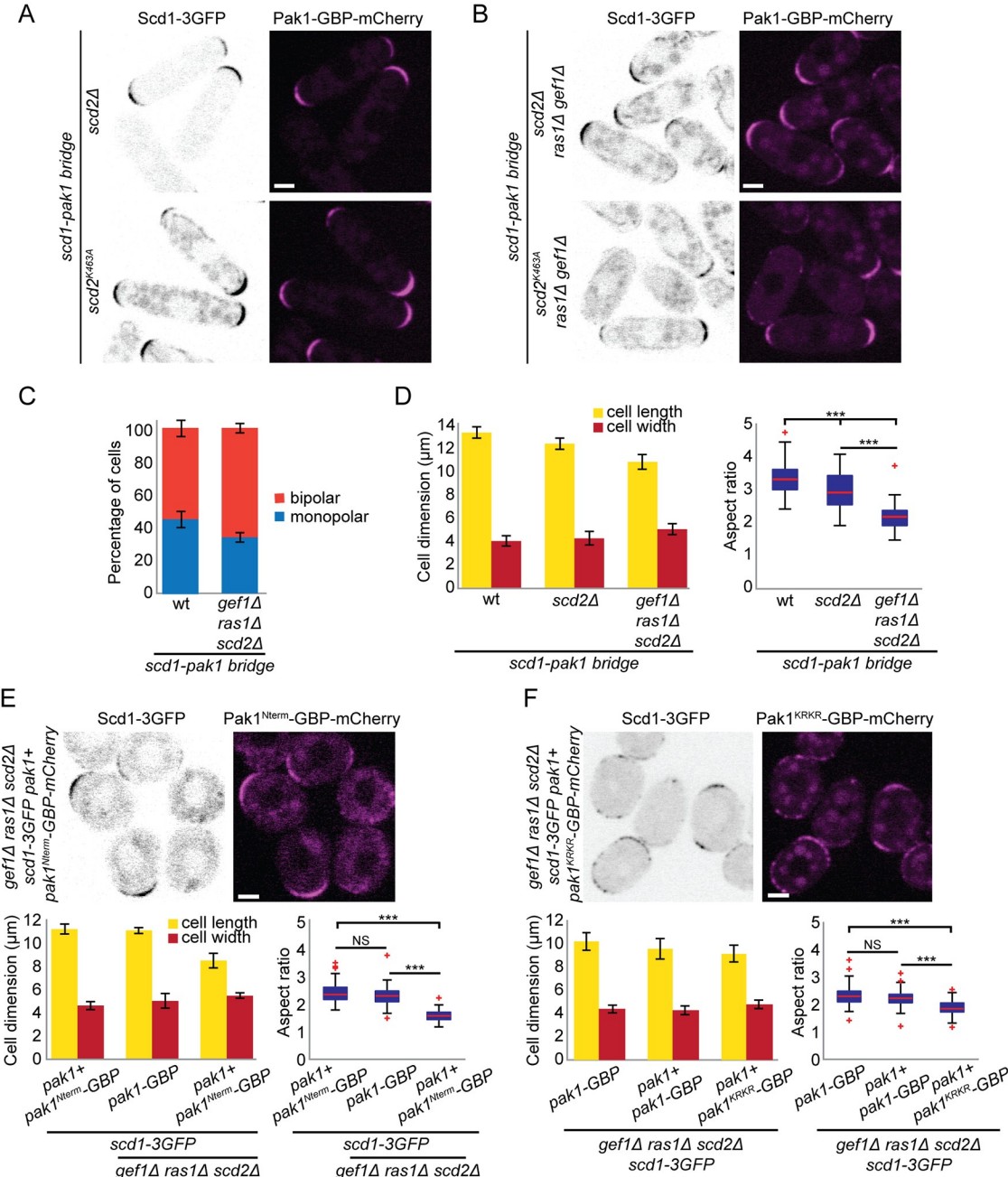

**Fig 6. An artificial Scd1-Pak1 bridge is sufficient to sustain bipolar growth.** (A-B) Localization of Scd1-3GFP (B/W inverted images) and Pak1-GBP-mCherry (magenta) in *scd2Δ* and *scd2^K463A* cells (A) or in *scd2Δ ras1Δ gef1Δ* and *scd2^K463A ras1Δ gef1Δ* cells (B). (C) Mean percentage of bipolar septated cells of strains with indicated genotypes. $N = 3$ experiments with $n > 30$ cells. (D) Mean cell length and width at division (left) and aspect ratio (right) of wt, *scd2Δ*, and *scd2Δ ras1Δ gef1Δ* strains expressing the *scd1-pak1 bridge*. $N = 3$ experiments with $n > 30$ cells. ***$3.5e^{-48} \leq p \leq 2e^{-7}$. (E-F) Localization of Scd1-3GFP (B/W inverted images) and Pak1^Nterm-GBP-mCherry (magenta) (E) and Scd1-3GFP (B/W inverted images) and Pak1^KRKR-GBP-mCherry (magenta) in *scd2Δ ras1Δ gef1Δ* strains (top) and mean cell length and width at division (bottom left), and aspect ratio (bottom right), of strains with indicated genotypes. $N = 3$ experiments with $n > 30$ cells. ***$1.72e^{-50} \leq p \leq 4.2e^{-44}$ (E) and $1.17e^{-27} \leq p \leq 5.8e^{-23}$ (F). Bar graph error bars show standard deviation; box plots indicate median, 25th and 75th percentiles, and most extreme data points not considering outliers, which are plotted individually using the red "+" symbol. Bars = 2 μm. All underlying numerical values are available in S6 Data. B/W, black and white; GBP, GFP-binding protein; NS, not significant; wt, wild type.

bridge. Analogous experiments bridging Pak1-GBP with Scd1 tagged with a single GFP also rescued the shape of *scd2Δ* and the viability of *scd2Δ gef1Δ ras1Δ* cells, which were, however, fairly round (Table 1 and S1 Table). This suggests that the strength, valency, and/or orientation of the Scd1-Pak interaction modulate the function of the bridge in promoting cell polarization. These data demonstrate that the constitutive association of Scd1 and Pak1 is sufficient to sustain viability and bipolar growth in the absence of other key Cdc42 regulators.

To further investigate the role of Pak1 kinase activity in feedback function, we bridged 2 different Pak1-GBP alleles to Scd1: an allele containing only the CRIB domain (Pak1$^{Nterm}$), responsible for binding Cdc42-GTP [38, 46, 49], and a kinase-dead version of Pak1 (Pak1$^{KRKR}$). Since Pak1 is essential for viability, these constructs were integrated at an ectopic genomic locus and expressed from the *pak1* promoter in addition to the endogenous wt *pak1+* gene. Both Scd1-Pak1$^{Nterm}$ and Scd1-Pak1$^{KRKR}$ bridges were able to support viability of *scd2Δ gef1Δ ras1Δ* cells, like a similarly expressed Scd1-Pak1$^{WT}$ bridge (Table 1, Fig 6E and 6F), indicating that bringing the GEF Scd1 in proximity to active Cdc42 through CRIB domain is sufficient for some level of life-sustaining feedback. By contrast, the Scd1-Pak1$^{Nterm}$ bridge did not restore rod shape, and the Scd1-Pak1$^{KRKR}$ bridge did so only partially (Fig 6E and 6F). By contrast, the Scd1-Pak1$^{WT}$ bridge restored rod shape whether expressed from the endogenous locus or in addition to endogenous Pak1. This suggests an important role for the Pak1 kinase domain.

## Ras1 promotes Cdc42 activation via Scd1

Because cells expressing the (full-length) Scd1-Pak1 bridge are rod-shaped, with only small variation in width, we reasoned that we could use these cells to probe Scd1 regulation without the confounding factor of cell shape change caused by absence of Scd2 and Ras1. To measure Cdc42 activity, we expressed CRIB-3mCherry under the control of the constitutive *act1* promoter in cells also carrying Scd1-3GFP and Pak1-GBP (Fig 7A). This allowed us to simultaneously measure the levels of Scd1 and Cdc42-GTP at cell poles. In all mutants tested, CRIB-3mCherry was localized at the cell tips as expected (Fig 7A, S9A Fig). All combinations carrying *ras1* deletion showed reduced levels of CRIB-3mCherry at the cell tips, as compared with *ras1+* strains, suggesting that Cdc42 activity strongly depends on Ras1 (Fig 7A–7C, S9 Fig). Importantly, comparing CRIB to Scd1 ratios at cell tips across mutant backgrounds clearly showed a lower ratio in *ras1Δ* cells (Fig 7C and 7D), indicating stronger effect of Ras1 on Cdc42 activity than would be predicted from its role in localizing the GEF Scd1 (Fig 1). For instance, whereas the absolute levels of Scd1 at cell tips were reduced to similar extent in *scd2Δ* and *ras1Δ*, CRIB levels were much lower in *ras1Δ*, with CRIB/Scd1 ratio reduced by more than 2-fold (Fig 7C). These data suggest that the role of Ras1 in Cdc42 activation goes beyond localizing Scd1.

To test whether Ras1 promotes Cdc42 activation by enhancing Scd1 activity or through alternative mechanisms, we compared CRIB cell pole levels between *scd1Δ* and *ras1Δ scd1Δ*, which showed indistinguishable very low CRIB levels (Fig 7E and 7F). These data suggest that Ras1 has no further effect in absence of Scd1, although it is possible that a small reduction would be missed due to the detection threshold of our assay. These data indicate that Ras1 promotes not only Scd1 localization (Fig 1) but also the strength of Scd1-mediated Cdc42 activation.

## Ras1 and Scd2-dependent positive feedback cooperate to ensure robust zones of Cdc42 activity

Although our data demonstrate that Ras1 is a critical regulator of Cdc42 activity by promoting Scd1 localization and activity, it is remarkable that constitutive activation of Ras1 has no morphological consequence during vegetative growth. Indeed, in *gap1Δ* cells, which lack the only

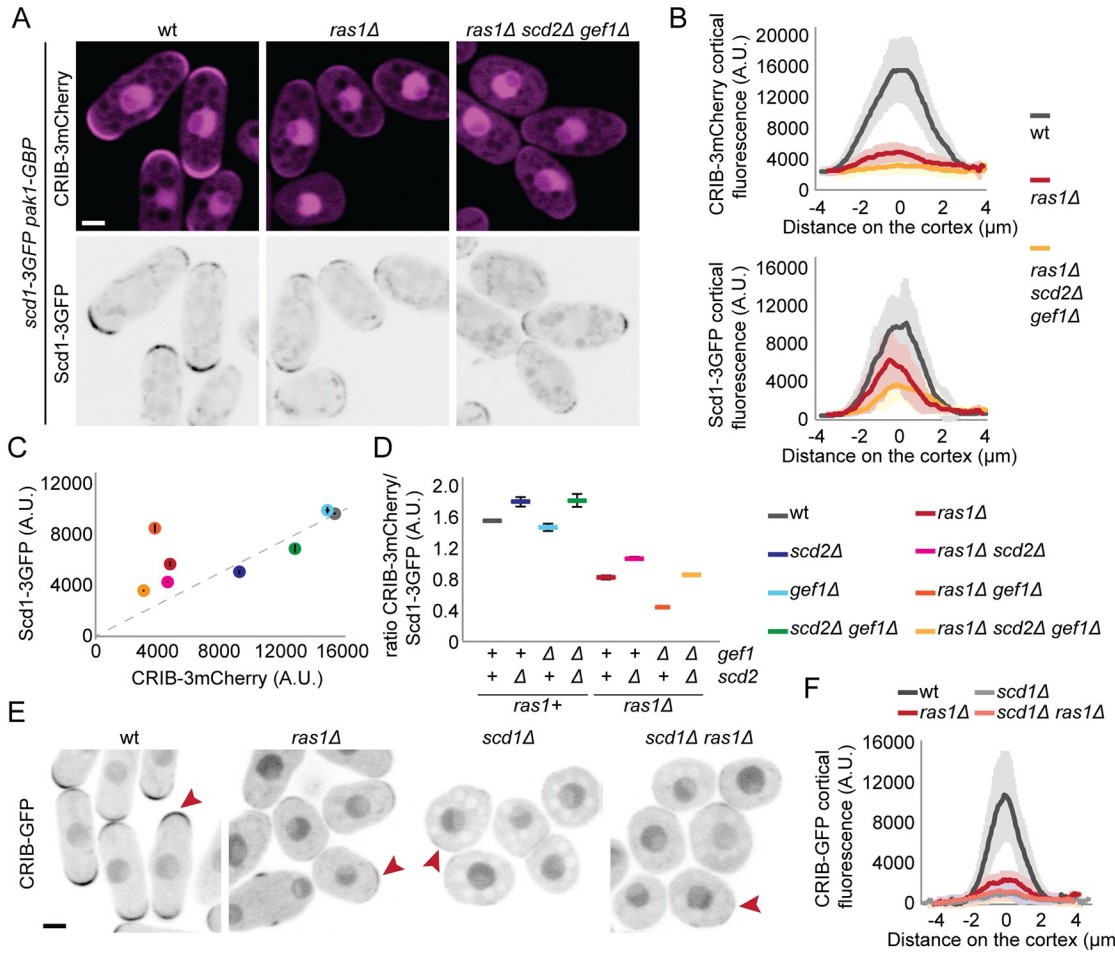

**Fig 7. Ras1 promotes the activation of the Cdc42 GEF Scd1.** (A) Localization of Scd1-3GFP (B/W inverted images) and CRIB-3mCherry (magenta) in wt, *ras1Δ* and *scd2Δ ras1Δ gef1Δ* cells expressing the *scd1-pak1 bridge*. (B) Cortical tip profiles of CRIB-3mCherry (top) and Scd1-3GFP (bottom) fluorescence in strains as in (A); *n* = 30 cells. (C) Plot of CRIB-3mCherry versus Scd1-3GFP fluorescence at the cell tip in strains of indicated genotypes expressing the *scd1-pak1* bridge and CRIB-3mCherry as in (A) and S9A Fig. (D) Ratio of CRIB-3mCherry and Scd1-3GFP fluorescence at the cell tip of strains as in (C). (E) Localization of CRIB-GFP in wt, *ras1Δ*, *scd1Δ*, and *ras1Δ scd1Δ* cells (B/W inverted images). (F) Cortical tip profiles of CRIB-GFP fluorescence in strains as in (E); *n* = 25 cells. In (B) and (F), thick line = average; shaded area = standard deviation. Bars = 2 μm. All underlying numerical values are available in S7 Data. A.U., arbitrary units; B/W, black and white; wt, wild type.

Ras1 GAP, Ras1 was active along the entire plasma membrane [39] (Fig 8A and 8B), but Scd1, Scd2, Gef1, and Cdc42, as well as Cdc42-GTP, all retained their wt localization patterns (Fig 8A–8D). To test whether this localization and cell shape depend on the described Scd1-Scd2-Pak1 positive feedback, we deleted *scd2* in cells lacking Gap1. Strikingly, *scd2Δ gap1Δ* double mutants were almost completely round and localized Scd1, CRIB, and Gef1 in large areas around the membrane (Fig 8D). Moreover, the expression of the Scd1-Pak1 bridge was sufficient to restore a rod morphology to these cells (Fig 8E and 8F), indicating that the scaffold-mediated positive feedback is sufficient to maintain Cdc42-GTP at the cell ends when Ras1-GTP does not convey spatial information. Thus, our results indicate that Scd2, by establishing a positive feedback on Cdc42, is important to constrain Cdc42 activity when Ras1 is constitutively activated.

Having established that the Scd1-Pak1–engineered positive feedback rescues the morphology of *scd2Δ gap1Δ* cells, we used this mutant to test whether the second GEF Gef1 can generate a positive feedback if bridged to Pak1. Surprisingly, recruitment of Gef1-3GFP to

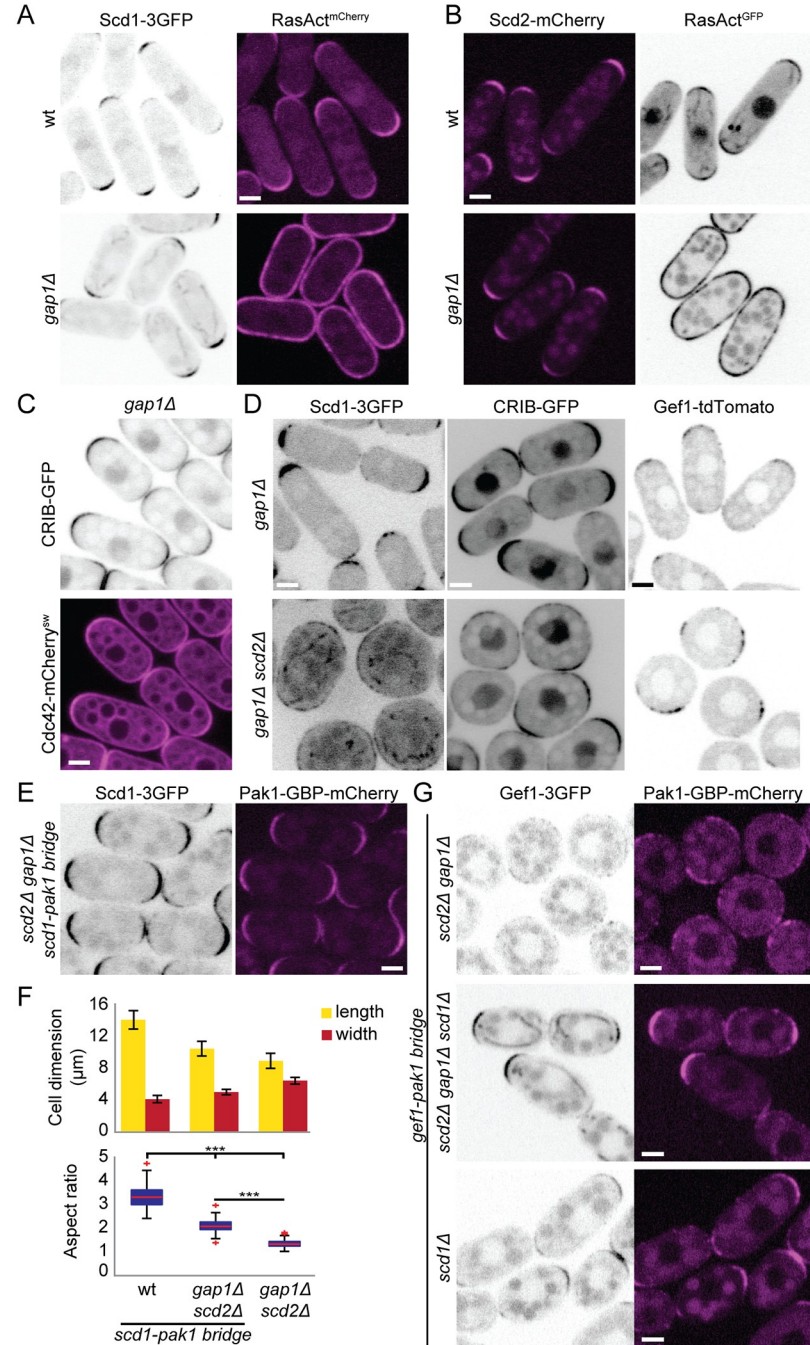

**Fig 8. Scaffold-mediated positive feedback restricts Cdc42 activity to cell tips when Ras1 activity is delocalized.**
(A-B) Localization of Scd1-3GFP (B/W inverted images) and RasAct$^{mCherry}$ (magenta) (A) or Scd2-mCherry
(magenta) and RasAct$^{GFP}$ (B/W inverted images) (B) in wt and $gap1\Delta$ cells. (C) Localization of Cdc42-mCherry$^{sw}$
(magenta) and CRIB-GFP (B/W inverted images) in $gap1\Delta$ cells. (D) Localization of Scd1-3GFP, CRIB-GFP, and
Gef1-tdTomato (B/W inverted images) in $gap1\Delta$ (top) and $gap1\Delta$ $scd2\Delta$ (bottom) cells. (E) Localization of Scd1-3GFP
(B/W inverted images) and Pak1-GBP-mCherry (magenta) in $gap1\Delta$ $scd2\Delta$ cells expressing the Scd1-Pak1 bridge. (F)
Mean cell length and width at division (top), and aspect ratio (bottom), of strains with indicated genotypes. $N = 3$
experiments with $n > 30$ cells. ***$9.55e^{-72} \leq p \leq 4.6e^{-52}$. (G) Localization of Gef1-3GFP (B/W inverted images) and
Pak1-GBP-mCherry (magenta) in $scd2\Delta$ $gap1\Delta$, $scd2\Delta$ $gap1\Delta$ $scd1\Delta$, and $scd1\Delta$ cells. Note that $scd2\Delta$ $gap1\Delta$ cells are
round, whereas $scd2\Delta$ $gap1\Delta$ $scd1\Delta$ cells are rod-shaped. Bar graph error bars show standard deviation; box plots
indicate the median, 25th and 75th percentiles, and most extreme data points not considering outliers, which are
plotted individually using the red "+" symbol. Bars = 2 μm. All underlying numerical values are available in S8 Data. B/
W, black and white; GBP, GFP-binding protein; wt, wild type.

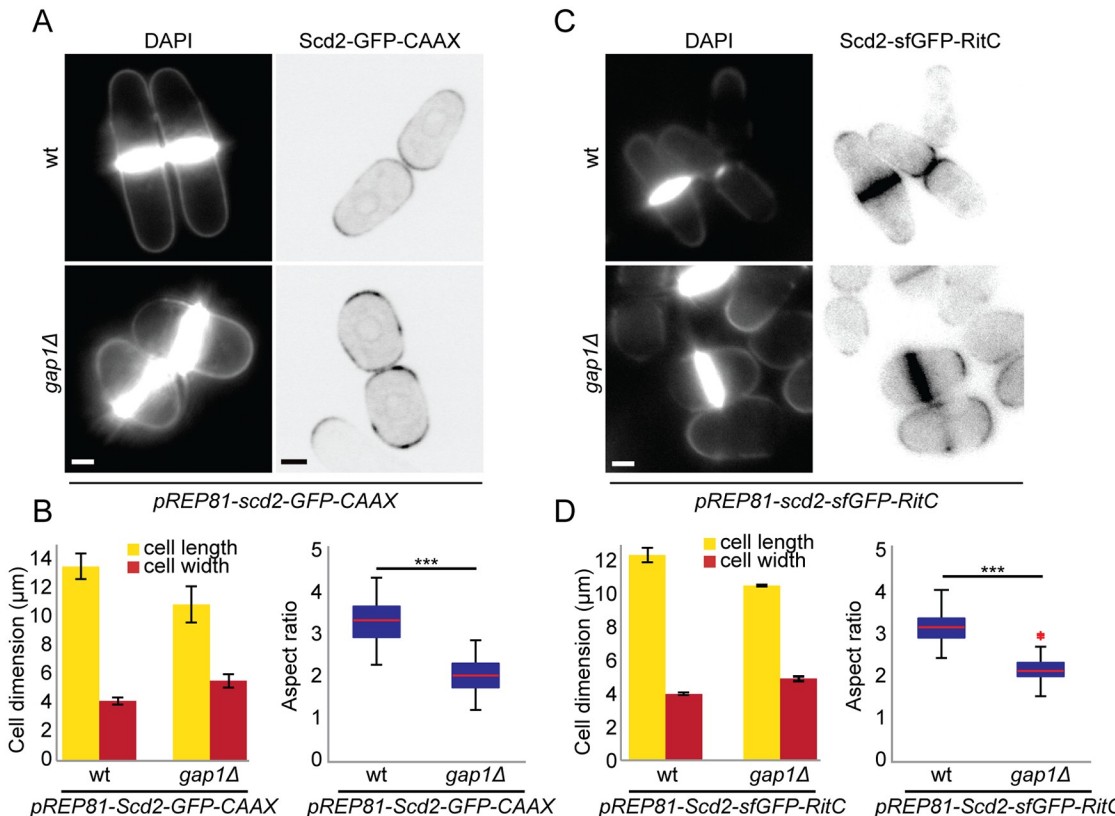

**Fig 9. Ras1 constrains Cdc42 activity when Scd2 is delocalized from cell tips.** (A) Calcofluor (left) and Scd2-GFP (right) images of wt and *gap1Δ* cells expressing *pREP81-scd2-GFP-CAAX* plasmid imaged 18 h after thiamine depletion for mild expression of Scd2-GFP-CAAX. (B) Mean cell length and width at division (left), and aspect ratio (right), of strains as in (A). $N = 3$ experiments with $n > 30$ cells. ***$p = 9.7e^{-45}$. (C) Calcofluor (left) and Scd2-sfGFP (right) images of wt and *gap1Δ* cells expressing *pREP81-scd2-sfGFP-RitC* plasmid imaged 48 h after thiamine depletion for mild expression of Scd2-sfGFP-RitC. (D) Mean cell length and width at division (left), and aspect ratio (right), of strains as in (C). $N = 3$ experiments with $n = 30$ cells. ***$p = 3.21e^{-50}$. Bar graph error bars show standard deviation; box plots indicate the median, 25th and 75th percentiles, and most extreme data points not considering outliers, which are plotted individually using the red "+" symbol. Bars = 2 μm. All underlying numerical values are available in S9 Data. wt, wild type.

Pak1-GBP-mCherry (Gef1-Pak1 bridge) did not rescue the morphological defects of *scd2Δ gap1Δ* cells (Fig 8G). This was not due to loss of function of Gef1 when forced in the proximity of Pak1, as the Gef1-Pak1 bridge as sole Gef1 copy not only was viable in *scd1Δ* cells but also promoted rod-shape formation (Table 1 and S1 Table; Fig 8G, bottom). We reasoned that the failure of the Gef1-Pak1 bridge to polarize the *scd2Δ gap1Δ* cells may be due to the presence of Scd1 GEF. Indeed, as shown in Fig 8D, Scd1 is no longer constrained by feedback regulation and is recruited and presumably activated by Ras1 over large areas at the cortex of *scd2Δ gap1Δ* cells, leading to large zones of Cdc42 activation. Remarkably, deletion of *scd1* from *scd2Δ gap1Δ* cells allowed the Gef1-Pak1 bridge to confer a rod shape to these cells (Fig 8G). We conclude that the Gef1-Pak1 bridge provides positive feedback but is unable to compete with Ras1-activated Scd1.

In complementary experiments, we tested whether locally restricted Ras1 activity is important when the positive feedback is not spatially restricted by targeting Scd2 to the plasma membrane through addition of a C-terminal prenylation signal or amphipathic helix. Expression of Scd2-CAAX or Scd2-RitC from a weak inducible promoter in addition to wt *scd2* led to extended localization of Scd2 along cell sides, in addition to cell poles (Fig 9A and 9C, compare

to Scd2 localization in Fig 8B), and resulted in rounding of *gap1Δ* but not wt cells (Fig 9B and 9D). This indicates that local Ras1 activity is important to maintain proper polarized growth when Scd2 is delocalized. Our results elucidate the requirement of a dual system for Cdc42 GTPase local activation through positive feedback and Ras1-dependent regulation: when 1 of the 2 systems is impaired or delocalized, the other becomes essential to restrict Cdc42 activity to cell tips.

## Discussion

Spatially restricted processes require polarity regulators to define discrete zones for effector protein recruitment and/or function. Feedback regulations are prevalent in cell polarization [3, 56, 57]. Positive- and negative-feedback loops within polarity modules may amplify or moderate input signals and cause dynamic competition and oscillations [23, 58]. Because Cdc42 drives cell polarization in distinct organisms, with qualitatively distinct cell shape outcomes, it is crucial to understand its feedback regulation across multiple model systems. Here, we implemented a novel optogenetic system to directly test positive feedback in Cdc42 regulation in fission yeast. We find that active Cdc42 binds the scaffold protein Scd2 to recruit its own activator, leading to activation and accumulation of additional Cdc42 molecules. By genetic approaches, we then deconstructed and reconstructed the positive feedback, which remarkably is sufficient to amplify positional signals and generate bipolar zones of Cdc42 activity in absence of other Cdc42 regulators. Cells with reengineered positive feedback further reveal that Ras1 GTPase plays a dual role: on one hand, it provides a positional input by recruiting the Cdc42 GEF Scd1; on the other, it modulates the strength of the positive feedback by enhancing Cdc42 activation through Scd1.

### Optogenetic manipulation of fission yeast proteins

A number of optogenetic systems, which harness the natural biology of photoreceptors to construct genetically encoded light-responsive dimerization modules, have been used to dissect signaling pathways [59, 60]. In particular, protein targeting through light stimulation provides an acute stimulus that differentiates proximal from distal effects of the protein of interest. Several light-inducible dimerization systems have been implemented in the budding yeast *S. cerevisiae* [61], but none had been used in *S. pombe*. Here, we establish the CRY2PHR-CIBN system [43], which is based on the *Arabidopsis* Cryptochrome 2 protein and its binding partner CIB1 [44], to manipulate protein localization in fission yeast cells. We note that we were unsuccessful in setting up 2 other optogenetic systems based on the light-induced binding of light oxygen voltage (LOV) domains to their natural or engineered binding partners, namely the TULIPs and iLID system [62, 63].

Similar to rates reported in other cells [43], the binding of CIBN to CRY2PHR is extremely fast after blue-light stimulation, with plasma-membrane recruitment occurring in seconds, and at blue-light dosages compatible with standard fluorescence imaging protocols. This allowed us to co-image the recruitment kinetics of endogenous GFP-tagged proteins. In principle, it would be possible to extend the system to simultaneously monitor 3 endogenous proteins—tagged with GFP, red fluorescent protein (RFP), and blue fluorescent protein (BFP) —by using untagged CRY2PHR and CIBN moieties. We note however that our system does not allow local protein recruitment. Indeed, since the photosensitive moiety (CRY2PHR) is cytosolic, where diffusion occurs at high rates, even local photoactivation leads to a global response in the small yeast cell. When we tried to invert the system, to constrain CRY2PHR diffusion to the plasma membrane by linking it to the RitC amphipathic helix, CRY2PHR was activated by light as it formed oligomers but failed to recruit its CIBN binding partner. The

CRY2PHR-CIBN system is thus ideally suited to induce a temporally acute relocalization and follow kinetic responses. We note that the light-dependent clustering properties of CRY2, which we have not characterized here, may also be exploited, as they have in other organisms [45, 47, 48].

### Positive feedback of Cdc42 activation

Studies of feedback regulation are complicated by circularity of the signal. Whereas recruitment of constitutively active GTP-bound Cdc42 to the cell cortex leads to cell rounding over time, the optogenetic approach provides an acute stimulus that permits monitoring the immediate cellular response to active Cdc42-GTP recruitment to the plasma membrane before any change in cell shape. Although the light-controlled constitutively active Cdc42 allele Opto$^{Q61L}$ may not provide normal levels of Cdc42 activity—for instance because of its mode of plasma-membrane interaction—this tool allowed us to show that Cdc42-GTP recruits its own GEF Scd1 in a manner that depends on direct binding to the Scd2 scaffold. In turn, this leads to recruitment of endogenous Cdc42, demonstrating positive feedback. We extrapolate that native Cdc42 uses the same mechanism of self-enhancement. Previous detailed in vitro work had shown that Scd2 directly binds both Pak1 and Cdc42-GTP and stimulates the interaction of Pak1 with Cdc42-GTP [46, 49]. Scd2 also directly binds Scd1 [37], likely through its PB1 domain, since the interaction is impaired by the K463A mutation [50]. In addition, Scd2, Scd1, Pak1, and Cdc42-GTP can form a quaternary complex in vitro [38]. Our results are entirely consistent with this analysis and indicate such complex also forms in vivo. Furthermore, forced complex formation by bridging Pak1 and Scd1 bypasses Scd2's function, indicating this is the main, if not sole, function of Scd2. Thus, by binding Scd1, Pak1, and Cdc42-GTP, Scd2 promotes positive-feedback enhancement of Cdc42 activity (Fig 10). The observation that clusters of Cdc42 at the cell cortex in the Opto$^{WT}$ system also recruit some Scd2, albeit more slowly than Opto$^{Q61L}$, suggests that Opto$^{WT}$ becomes active because Scd2 recruitment strictly depends on Cdc42 activity. This suggests the interesting possibility that enhanced proximity of Cdc42 may be sufficient to trigger the feedback, perhaps upon

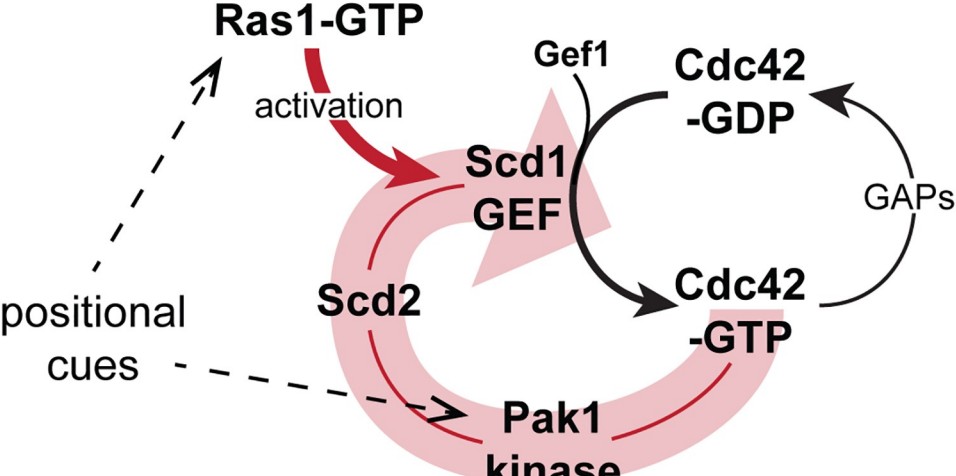

**Fig 10. Model of scaffold-mediated positive-feedback regulation of Cdc42 activity gated by Ras1 GTPase.** The schematics shows the Scd2 scaffold-mediated positive feedback on Cdc42 activation (wide, pale red arrow) and input by Ras1-GTP. By promoting Scd1 activity, Ras1-GTP amplifies the positive feedback. Positional cues on Cdc42 are transmitted through both Ras1-GTP and Pak1. GAP, GTPase activating protein; GEF, guanine nucleotide exchange factor.

stochastic Cdc42 activation. The Opto$^{WT}$ activity is, however, not sufficient to cause morphological changes, perhaps because absence of Ras1 and presence of Cdc42 GAP proteins on cell sides prevent full activation and competition with natural polarity sites at cell poles.

Our observations mirror similar ones made in *S. cerevisiae*, in which the scaffold Bem1 (Scd2 homologue) promotes the formation of a Cdc42-GEF-PAK complex [27, 29, 33]. Reconstitution of this positive feedback is sufficient to break symmetry in absence of the scaffold and positional cues provided by the Ras-like GTPase Rsr1 [28]. The feedback also promotes sustained Cdc42 activation in late G1, for which optogenetic strategies have also been used to demonstrate that scaffold and GEF induce their own recruitment [35]. In conjunction with competition between polarity patches, the positive feedback is further thought to promote a winner-take-all outcome, yielding a single patch of Cdc42 activity [30–32], a phenomenon also observed in the optogenetic study above [35]. The finding that an equivalent feedback exists in fission yeast cells shows that this positive feedback has been conserved at least since the diversification of ascomycetes and likely exists in all species expressing an Scd2/Bem1 scaffold homologue, which is present throughout fungi. Interestingly, our finding that artificial reconstitution of the feedback by simply linking the Scd1 GEF and PAK leads to bipolar growth also demonstrates that singularity or bipolarity are not directly determined by the scaffolding function itself.

The positive feedback described above specifically relies on the GEF Scd1. Indeed, Gef1 is not part of the feedback and provides an additive function to Scd1, as it is essential for cell viability in absence of Scd1 or its regulators (this work; [11, 12]). In these conditions, Gef1 strongly colocalized with dynamic zones of Cdc42 activity at the cell cortex, which maintain the cell alive but do not sustain polarized growth. This is different from the situation in wt cells, in which Gef1 is largely cytosolic [15]. Interestingly, *gef1Δ* cells are defective in bipolar growth initiation [11], yet cells expressing the Scd1-Pak1 bridge are bipolar even in absence of Gef1. One possibility is that Gef1 serves as enhancer to stochastic Cdc42 activation, such that Cdc42-GTP can act as seed for the positive feedback, which then promotes the stabilization of the polarity axis. This hypothesis would be consistent with recent data suggesting a priming function of Gef1 in some conditions [64]. Although Gef1 is not naturally part of the positive feedback, we could force it to act within the positive feedback by bridging it to Pak1 and largely replace Scd1's role in morphogenesis. These data suggest that the key contributions of the GEF protein in the positive feedback are the GEF activity itself and interaction with the Cdc42 effector. The ability to substitute one GEF for the other in the synthetic positive feedback raises the question of whether and how negative feedback, which is proposed to operate through phosphorylation of the GEF by the PAK in budding yeast, may contribute. Likewise, whether and how Ras1 regulates the activity of the Gef1-Pak1 bridge will require further investigation.

## The Pak1 effector provides positional information to Cdc42

The polarized growth of cells bearing a synthetic Scd1-Pak1 or Gef1-Pak1 bridge raises the question of how the Cdc42 core module receives positional information that anchors it at the cell tips. Linking the GEF with Pak1 through GFP-GBP binding does not per se provide any positional signal, and yet it restores bipolar growth to cells lacking Scd2, Ras1, and Gef1. Thus, none of these 3 proteins provide an essential positional cue, which in consequence must be conferred by the GEF and/or Pak1. Since Scd1 localization itself strictly depends on Scd2 and Ras1, which are themselves dispensable, it follows that positional information must be conferred by the Cdc42 substrate Pak1 (Fig 10).

This conclusion adds to the increasing number of functions for PAK-family kinases in feedback regulation of Cdc42. Pak1 is an integral component of the scaffold-mediated complex

promoting positive feedback. PAK activity is strongly implicated in negative-feedback regulation in both *S. pombe* and *S. cerevisiae*, at least in part through phosphorylation of the GEF [13, 33, 34]. Our work now establishes that Pak1 also provides spatial information to position sites of Cdc42 activity, which is likely reinforced through feedback regulation. Our findings that restoring feedback with only Pak1 CRIB domain leads to viable but unpolarized cells is consistent with the idea that Pak1 contains positional information beyond the CRIB domain. As Scd1 recruitment through a kinase-dead version of Pak1 also impedes efficient restoration of polarized growth, and PAK activity has been involved in negative-feedback regulation [13, 33, 34], the compromised function of this construct may also stem from lack of negative feedback. The specific roles of Pak1 kinase domain and activity remain open questions for the future.

Which specific positional signals are read by Pak1 remain to be established. One possibility is that Pak1 reads information provided by microtubules, which deposit the polarity factors Tea1, Tea3, and Tea4 at the cell cortex [55, 65–67]. For instance, Pak1 phosphorylates Tea1 [68] and the related protein Tea3, which then competes with Scd2 for Pak1 binding [69]. Microtubules also contribute positional information to Gef1-mediated Cdc42 activation [15]. The actin cytoskeleton has also been proposed to play important yet controversial roles in Cdc42 feedback regulation [23]. It may indeed play some role, as it is linked to Tea4 through the formin For3 [65]. However, a previously assumed major role of the actin cytoskeleton, which was revealed by the displacement of the Cdc42 feedback module from cell poles upon depolymerization of F-actin [6, 70], is in fact a consequence of mitogen activated protein kinase (MAPK) stress signaling rather than a direct effect on protein recruitment by the actin cytoskeleton [71]. The Scd1-Pak1-bridged cells lacking all other Cdc42 regulators now offer a simplified system to further dissect these positional signals.

### Ras1 promotes the localization and activity of the Cdc42 GEF Scd1

Although the increased diameter of *ras1Δ* cells complicates analysis of Ras1 function, it is clear that Ras1 is required for efficient Scd1 localization to cell poles. Because of the physical interaction of Ras1 with Scd1 [37], this suggests a direct function of Ras1 on Scd1. The question is whether Ras1 confers positional information by recruiting Scd1 to cell poles and/or serves to activate Scd1 and thus indirectly promotes Scd1 localization by potentiating the positive feedback. Our data indicate that Ras1 has both functions (Fig 10). First, our data provide evidence that Ras1 potentiates the feedback by promoting the activation of Cdc42 (Fig 10). Indeed, using the fairly invariant shape of cells containing the Scd1-Pak1 bridge, independent of the presence of Scd2, Ras1, and Gef1, we found that *ras1Δ* mutants consistently showed reduced Cdc42-GTP/Scd1 ratios at cell tips, indicating exaggerated reduction in Cdc42 activity. This is in fact also visible in cells lacking the Scd1-Pak1 bridge: *ras1Δ* and *scd2Δ* mutants display similar Scd1 levels, but *ras1Δ* cells have only half of the CRIB signal (see Fig 1B). Because deletion of Ras1 leads to strong reduction in Cdc42 activity in *scd1+* but not *scd1Δ* cells, we conclude that Ras1 promotes Cdc42 activation by enhancing Scd1 activity. Ras1 may promote Scd1 activity directly, which would be in line with Ras1 directly binding Scd1 [37]. The view that Ras1 promotes Scd1 activity is also consistent with the finding that Cdc42 activation over large areas of the cortex by constitutively active Ras1 in *gap1Δ* cells depends on Scd1. We conclude that Ras1 promotes Cdc42 activation by Scd1 GEF, thus potentiating the feedback.

Because positive-feedback regulation reinforces the localization of the Scd1-Scd2-Cdc42-Pak1 complex, by promoting Scd1 activity, Ras1 also enhances its localization. However, this is not the sole mechanism by which Ras1 promotes Scd1 localization. Indeed, in *scd2Δ* cells that lack the scaffold-mediated feedback, we find that Ras1-GTP plays an instructive role in the localization of Scd1: in absence of Ras1, Scd1 completely fails to localize to the cortex; in *ras1+*

cells, where Ras1 activity is restricted to cell poles by yet-to-be-defined cues, Scd1 localizes to cell poles; in *gap1Δ* cells where Ras1 is uniformly active, Scd1 occupies large areas of the cell cortex. Thus, Ras1 also directs Scd1 localization independently of the scaffold-mediated positive feedback. Consistent with Ras1 acting independently of the feedback, we did not detect Ras1-GTP at the cell sides after light-induced relocalization of Opto$^{Q61L}$ to the membrane. The persistent cell-tip localization of Scd1 in *gap1Δ* cells, which exhibit uniform Ras1-GTP distribution and intact positive feedback, suggests that Pak1 positional cues are dominant. This is likely due to nonlinear amplification through the feedback leading to a winner-takes-all behavior, whereas the Ras1 input may be linear. In summary, the data described herein establish that Ras1 not only promotes Scd1 activity but also provides positional information to localize it, a second function that is particularly important in absence of positive feedback.

Our data are consistent with a proposed role of the Ras-like GTPase Rsr1 in *S. cerevisiae*. Rsr1-GTP, a key player in bud site placement [72], directly binds and recruits the Cdc42 GEF [73–75]. Although Rsr1 is widely believed to play a positional role, it appears to have an additional function in symmetry breaking beyond its role in bud site selection [76], which may involve activation of the GEF Cdc24 [77]. In mammalian cells, Ras GTPase also promotes GEF activity toward the Cdc42-related Rac GTPase [78]. Thus, activation of Rac/Cdc42 GEFs might be an evolutionarily conserved function of Ras-family GTPases.

### Dual control of Cdc42 by positive feedback and Ras1 confers robustness

The observation that Cdc42 activation by Scd1 is under both positive feedback and Ras1 regulation raises the question of how these two modules cooperate. Our results suggest that by modulating the GEF capacity of Scd1, active Ras1-GTP delineates cortical zones in which the Cdc42 feedback operates efficiently. Indeed, the growth of *gap1Δ* cells that constitutively activate Ras1 is perturbed by changes in the Cdc42 feedback due to either increase or removal of the Scd2 scaffold. In a similar manner, Scd1 overexpression causes cell rounding of *gap1Δ* mutants but not wt cells, indicating that cells lacking spatial Ras1-dependent cues become sensitive to perturbations in feedback control. The importance of a tight coupling between the Ras1 input and the positive feedback is evident from the poor morphogenesis of *scd2Δ gap1Δ* cells that express the Gef1-Pak1 bridge, in which positive feedback operates through the GEF Gef1, and Ras1 functions through GEF Scd1. These results suggest that cell polarization is best achieved upon coupling of Ras1 and feedback controls through a single GEF. Ras1 modulation of the positive feedback might be particularly relevant during sexual reproduction for partner selection and mating, when cells abolish tip growth and pheromone signaling positions active Ras1 [39, 42]. We propose that by superimposing positive feedback and Ras1-dependent regulation, cells buffer fluctuations of individual regulators to robustly define and position zones of Cdc42 activity.

In summary, the formation of a complex between a Cdc42 activator and PAK effector underlies an evolutionarily conserved positive feedback that amplifies weak positional cues for cell polarization. This positive feedback is further gated and amplified through activation of the GEF Scd1 by Ras1 GTPase, which also provides positional information. Thus, 2 complementary mechanisms synergize to yield robust sites of Cdc42 activity.

## Methods

### Strains, media, and growth conditions

Strains used in this study are listed in S2 Table. Standard genetic manipulation methods for *S. pombe* transformation and tetrad dissection were used. For microscopy experiments, cells were grown at 25 °C in Edinburgh minimal medium (EMM) supplemented with amino acids as required. For optogenetic experiments (Figs 2–5 and S2–S7 Figs), cells were first precultured

in 3 mL of EMM in dark conditions at 30 ˚C for 6–8 h. Once exponentially growing, precultures were diluted (optical density [O.D.]$_{600nm}$ = 0.02) in 10 mL of EMM and incubated in dark conditions overnight at 30 ˚C. In order to allow proper aeration of the culture, 50 mL Erlenmeyer flasks were used. All live-cell imaging was performed on EMM-ALU agarose pads. Gene tagging was performed at endogenous genomic locus at the 3′ end, yielding C-terminally tagged proteins, as described [79]. Pak1 gene tagging was performed by transforming a wt strain with AfeI-linearized pBSII(KS$^+$)-based single integration vector (pAV72-3′UTR$^{pak1}$-Pak1-sfGFP-kanMX-5′UTR$^{pak1}$) targeting the endogenous locus. The functional mCherry-tagged and sfGFP-tagged Cdc42 alleles Cdc42-mCherry$^{sw}$ and Cdc42-sfGFP$^{sw}$ were used as described in [6]. RasAct$^{GFP}$ and RasAct$^{mCherry}$ probes to detect Ras1 activity were used as described in [39]. Gene deletion was performed as described [79]. Gene tagging, gene deletion, and plasmid integration were confirmed by diagnostic PCR for both sides of the gene.

Construction of a strain expressing CIBN-mTagBFP2-Ritc was done by integration at the *ura4* locus of the pSM2284 plasmids linearized with AfeI. All inserts were generated by PCR, using Phusion high-fidelity DNA polymerase (New England Biolabs, Ipswich, MA, USA) according to manufacturer's protocol and cloned into pBSII(KS$^+$)-based single integration vector (pAV0133). *pTDH1* was amplified from wt genomic DNA (gDNA) using osm3758 (5′-tcc*ggtacc*gggcccgctagcatgcTAAAGTATGGAAAATCAAAA-3′, consecutive KpnI-ApaI-SphI restriction enzyme sites) and osm3759 (5′-tcc*ctcgag*tacgtaTTTGAATCAAGTGTAAATCA-3′, consecutive XhoI-SnaBI restriction enzyme sites) and cloned with KpnI-XhoI. *CIBN* was amplified from pSM1507 (Addgene ID: 26867, [43]) using osm3179 (5′-tcc*gtcgac*CATGAATG GAGCTATAGGAGG-3′) and osm3180 (5′-tcc*cccgggg*cATGAATATAATCCGTTTTCTC-3′) and cloned with SalI (ligation with XhoI)-XmaI. *mTagBFP2* fluorophore was amplified from pSM1858 (Addgene ID: 54572) using osm4466 (5′-*accggt*cgccaaATGGTGTCTAAGGGCGA AGA-3′) and osm4467 (5′-tcc*gtcgac*ATTAAGCTTGTGCCCCAGTT-3′) and cloned using AgeI-SalI. *Ritc* was amplified from pSM1440 [6] using osm2893 (5′-tcc*gtcgac*CACAAGAAAA AGTCAAAGTGTC-3′) and osm2894 (5′-tcc*gcggccgc***TCA**AGTTACTGAATCTTTCTTC-3′) and cloned with SalI-NotI. *ScAdh1*$_{terminator}$ was amplified from wt *S. cerevisiae* gDNA using osm3131 (5′-tcc*gcggccgc*ACTTCTAAATAAGCGAATTTC-3′) and osm3144 (5′-tcc*gagctc*tctg catgcATATTACCCTGTTATCCCTAG-3′, consecutive SacI-SphI restriction enzyme sites) and cloned with NotI-SacI. Finally, the plasmid was linearized with AfeI and integrated at the *ura4* locus to generate strain YSM3563.

To generate the Opto system, we combined the *pTDH1-CIBN-mTagBFP2-Ritc-ScADH1*$_{terminator}$ with *pAct1-CRY2PHR-mCherry* into a pBSII(KS$^+$)-based single integration vector. *pAct1* was amplified from wt gDNA using osm3750 (5′-tcc*ggtaccgagcatg*CGATCTAC GATAATGAGACGG-3′, consecutive KpnI-SphI restriction enzyme sites) and osm2379 (5′-cc*ggctcgag*GGTCTTGTCTTTTGAGGGT-3′) and cloned with SphI-XhoI. CRY2PHR-mCherry was amplified from psm1506 (Addgene ID: 26866) using osm2557 (5′- tcc*gtcgactg*ATCAATG AAGATGGACAAAAAGAC-3′, consecutive SalI-BclII restriction enzyme sites) and osm2559 (5′-tc*gcggccgc***TTA**tggcgcgccCTTGTACAGCTCGTCCATGCC-3′, AscI upstream to STOP codon followed by NotI site) and cloned with SalI-NotI. Finally, the plasmid pSM2287 was linearized with AfeI and integrated at the *ura4* locus to generate strain YSM3565.

To generate the Opto$^{Q61L}$ and Opto$^{WT}$ systems, we combined Cdc42$^{Q61L}$ΔCaaX and Cdc42$^{WT}$ΔCaaX with the Opto system respectively. Cdc42$^{Q61L}$ΔCaaX was amplified from pSM1354 using osm2909 (5′-tcc*ggcgcgcc*aATGCCCACCATTAAGTGTGTC-3′) and osm2975 (5′-tcc*gagctc*TTACTTTGACTTTTTCTTGTGAGG-3′) and cloned into pSM2287 with AscI-SacI. Cdc42$^{WT}$ΔCaaX was amplified from wt gDNA using osm2909 (5′-tcc*ggcgcgcc*aATGCC CACCATTAAGTGTGTC-3′) and osm2975 (5′-tcc*gagctc*TTACTTTGACTTTTTCTTGTG AGG-3′) and cloned with AscI-SacI. Note that Opto$^{WT}$ system was combined with a CIBN-GFP

variant constructed as mentioned before. Finally, the plasmids pSM2285 and pSM2286 were linearized with AfeI and integrated at the ura4 locus to generate strains YSM3566 and YSMXXX. The Opto$^{T17N}$ plasmid (pSM2593) was generated by performing site-directed mutagenesis PCR on pSM2286 using osm6933 (5′-CTGACCGTTTGAATGTAG**AAC**TGTCTGCTTATTTCC-3′) and osm6934 (5′-GGAAATAAGCAGACA**GTT**CTACATTCAAACGGTCAG-3′). By linearizing pSM2593 using AfeI and integrating it at the ura4 locus, strain YSM3688 was generated.

Construction of strains expressing different *scd2* alleles in Fig 5 and S7 Fig was done by integration at the endogenous *scd2* locus of the following plasmids linearized with AfeI: *pFA6a-3′UTR-AfeI-5′UTR-scd2-natMX-3′UTR* (pSM2263), *pFA6a-3′UTR-AfeI-5′UTR-scd2$^{K463A}$-natMX-3′UTR* (pSM2268), *pFA6a-3′UTR-AfeI-5′UTR-scd2$^{275-536}$-natMX-3′UTR* (pSM2272), *pFA6a-3′UTR-AfeI-5′UTR-scd$^{1-266}$-natMX-3′UTR* (pSM2302), *pFA6a-3′UTR-AfeI-5′UTR-scd2-eGFP-kanMX-3′UTR* (pSM2256), *pFA6a-3′UTR-AfeI-5′UTR-scd2$^{K463A}$-eGFP-kanMX-3′UTR* (pSM2262), *pFA6a-3′UTR-AfeI-5′UTR-scd2$^{275-536}$-eGFP-kanMX-3′UTR* (pSM2270), *pFA6a-3′UTR-AfeI-5′UTR-scd2$^{1-266}$-eGFP-kanMX-3′UTR* (pSM2306). First, a *pFA6a-3′UTR-AfeI-5′UTR-scd2-kanMX-3′UTR* (pSM2255) plasmid was generated by InFusion cloning (Clontech) of a pFA6a-based plasmid containing the yeast kanMX resistance cassette digested with KpnI and AscI, *scd2* 5′UTR amplified from wt gDNA with primers osm5687 (5′-gctCAGCAGTTCAGTCAC) and osm5688 (5′-GAAGCATACCTTTAACATCTCGAGAGAGACTGGAATTAGAAC), *scd2* 3′UTR amplified from wt gDNA with primers osm5685 (5′-CTGCAGGTCGAG<u>GGTACC</u>GACTATGTATATTTAAAG) and osm5686 (5′-GTGACTGAACTGCTGAGCGCTGATTAAGACGTTGTCAAGAAATG), and *scd2* ORF (STOP included) amplified from wt gDNA with primers osm5689 (5′-<u>CTCGAG</u>ATGTTAAAGGTATGCTTC) and osm5690 (5'-CTTATTTAGAAGTGGCGCGCCTCAAAACCTCCGTCTTTC). *pFA6a-3′UTR-AfeI-5′UTR-scd2-eGFP-kanMX-3′UTR* (pSM2256) plasmid was generated by InFusion cloning of pAV63 digested with KpnI and AscI, *scd2* 5′UTR amplified from wt gDNA with primers osm5687 and osm5688, *scd2* 3′UTR amplified from wt gDNA with primers osm5685 and osm5686, *scd2* ORF (no STOP) amplified from wt gDNA with primers osm5689 and osm5691 (5′-CTCGAGATGTTAAAGGTATGCTTC), and eGFP fragment amplified from pSM1080 (*pFA6a-eGFP-natMX*) with primers odm5692 (5′-CGGATCCCCGGGGTTAATTAACAG) and osm5693 (5′-CTTATTTAGAAGTGGCGCGCCCTATTTGTATAGTTCATC). Plasmids expressing the *scd2$^{K463A}$* mutation were obtained by site-directed mutagenesis with primers osm3432 (5′-GTTCGACATGCAAAGTT**GCA**GTCAGATTAGGAGATG) and osm3433 (5′-CATCTCCTAATCTGAC**TGC**AACTTTGCATGTCGAAC) on wt plasmids. To obtain plasmids expressing *scd2$^{aa275-536}$*, the *scd2$^{275-536}$* fragment was amplified from wt gDNA with primers osm5766 (5′-tcc<u>CTCGAGATG</u>ctgcaaacattggagtcgcgtacg) and osm5690 to obtain the untagged plasmid or osm5766 and osm5691 to obtain the eGFP-tagged plasmid, digested with XhoI and AscI and cloned in similarly treated wt plasmids. To obtain plasmids expressing *scd2 scd2$^{aa1-266}$*, the *scd2$^{1-266}$* fragment was amplified from wt gDNA with primers osm5689 and osm5764 (5′-<u>GGCGCGCC</u>***tca***ggaaccaggaaaagtgcttgaatt) to obtain the untagged plasmid or osm5689 and osm5765 (5′-A<u>CCCGGG</u>GATCCGggaaccaggaaaagtgcttgaatt) to obtain the eGFP-tagged plasmid, digested with XhoI and AscI or XhoI and XmaI and cloned in similarly treated wt plasmids. To obtain natMX plasmids, the natMX fragment was digested from pSM646 (*pFA6a-natMX*) with BglII and PmeI and cloned in similarly treated kanMX plasmids.

Construction of strains in Fig 6 expressing Pak1$^{N-term}$-GBP-mCherry (aa2–185), Pak1$^{wt}$-GBP-mCherry and Pak1$^{KRKR}$-GBP-mCherry (K418R, K419R) was done by integration of constructs under *pak1* promoter at the ura4+ locus. Expression of *pak1$^{N-term}$* allele was driven by the 630-bp sequence upstream of the *pak1* START codon followed by the START codon and 2 Gly codons amplified with primers osm2475 (5′-tcc<u>gtcgac</u>TCAAATTCACTGATTTAAGAC)

and osm2476 (5′-tcc**ccccggg**ACCTCCCATAGTAAATAAATTTATTAA) and cloned with SalI and XmaI. The *pak1*$^{N-term}$ fragment encoding aa2–185 was amplified using primers osm2477 (5′-tcc**ccccggg**GAAAGAGGGACTTTACAACC) and osm2479 (5′-tcc**ttaattaa**TGTAATGCCACTGACTTTTAG) and cloned with XmaI and PacI in frame with the GBP-mCherry sequence obtained from pAV52 (*pJK210-GBP-mCherry*, [80]), which was amplified with primers osm3329 (tccttaattaaCATGGCCGATGTGCAGCTGGTGG) and osm3331(cccggcgcgccttaCTTGTACAGCTCGTCCATGC). The fragments were cloned into a vector targeting the ura4 locus and carrying bleMX6 resistance cassette to obtain the plasmid *pura4-P*$^{pak1}$*-pak1*$^{N-term}$*-GBP-mCherry-bleMX-ura4+* (pAV273). Expression of *pak1*$^{WT}$ allele was driven by the 630-bp sequence upstream of the *pak1* START codon and first 3 codons amplified with primers osm2475 (5′-tcc**gtcgac**TCAAATTCACTGATTTAAGAC) and osm2693 (5′-tcc**gtcgac**CCCTCTTTCCATAGTAAATAA) and cloned by using the SalI restriction enzyme site. The *pak1*$^{WT}$ fragment was amplified with primers osm2700 (5′-tcc**gtcgac**GAAAGAGGGACTTTACAACCT) and osm2701 (5′-tcc**ccccggg**ccTTTACCAGAATGATGTATGGA) and cloned with SalI and XmaI in frame to GBP-mCherry sequence to obtain plasmid *pura4-P*$^{pak1}$*-pak1*$^{wt}$*-GBP-mCherry-bleMX-ura4+* (pAV558). *pak1* mutagenesis was carried out by site-directed mutagenesis with primers osm3682 (5′- CTAATCTTTCTGTTGCCATC**AGGAGA**ATGAACATTAATCAACAGCC) and osm3683 (5′-GGCTGTTGATTAATGTTCAT**TCTCCT**GATGGCAACAGAAAGATTAG) to obtain plasmid *pura4-P*$^{pak1}$*-pak1*$^{KRKR}$*-GBP-mCherry-bleMX-ura4+* (pAV559). Plasmids pAV273, pAV558, and pAV559 digested with AfeI were stably integrated as a single copy at the *ura4+* locus in the yeast genome.

Construction of strains in Fig 7 and S9 Fig expressing CRIB-3mCherry was done by integration of CRIB-3mCherry under *act1* promoter at the *ura4+* locus. First, *3xmCherry* fragment was digested with PacI and AscI from pSM2060 (*pFA6a-3xmCherry-nat*; kindly received from Ken Sawin, Edinburgh University; [71]) and cloned in pSM1822 (*pJK148-P*$^{pak1}$*-CRIB[gic2aa2-181]-mCherry-leu1+*) to generate plasmid *pJK148-P*$^{pak1}$*-CRIB(gic2aa2-181)-3xmCherry-leu1+* (pSM2095). Second, *P*$^{pak1}$*-CRIB(gic2aa2-181)-3xmCherry* fragment was digested with KpnI and NotI from pSM2095 and ligated to similarly treated pJK211 to generate plasmid *pura4-Ppak1-CRIB(gic2aa2-181)-3xmCherry-ura4+* (pSM2104). Third, *bsd* fragment was digested from pSM2081 (*pFA6a-bsd*) with AscI and SacI and cloned into similarly treated pSM2104 to generate plasmid *pura4-Ppak1-CRIB(gic2aa2-181)-3xmCherry-bsd-ura4+* (pSM2131). Fourth, the *act1* promoter was amplified with primers osm5921 (5′- ccg**ctcgag**GATCTACGATAATGAGACGGTGTTTG) and osm5922 (5′- tcc**CCCGGG**ACCTCCCATGGTCTTGTCTTTTGAGGGTTTTTTGG), digested with XmaI and XhoI and ligated into similarly treated pSM2131 to generate plasmid *pura4-P*$^{act1}$*-CRIB(gic2aa2-181)-3xmCherry-bsd-ura4+* (pSM2358). Finally, pSM2358 digested with AfeI was stably integrated as a single copy at the *ura4+* locus in the yeast genome.

The pSM1779 plasmid was used to express the Scd2-GFP-CAAX from the weak p$^{nmt81}$ promoter. We first amplified Scd2-GFP from the genome of YLM44 *scd2-GFP-kanMX* strain, which itself was obtained by tagging by PCR-based targeting using primers osm1359 (5′-CTTAATAATGTTGACGATTTACGGAAGGCATGTTCTCAAGAATCAGGAGTTTTACTTTTTGCAGAAAGACGGAGGTTTCGGATCCCCGGGTTAATTAA-3′) and osm1360 (5′-ACCATGAAAGAACGAAACGAAAAAAAAATAAAATGCAAGAACGTAATAAGAAACCCAAATCTTTAAATATACATAGTCGAATTCGAGCTCGTTTAAAC-3′) on the plasmid pFA6a-GFP-kan ([79]; pSM674). The Scd2-GFP fragment was amplified using primers osm3337 (5′- TCCCATATGTTAAAGATTAAAAGGACTTGGAAAAC-3′) and osm207 (5′-cag**cccggg**TTATGAAATGATGCATTTGTATAGTTCATCCATGCC-3′) and cloned into pREP81 vector [81] using NdeI and XmaI restriction sites. Next, we used the XmaI restriction sites to replace the GFP with the GFP-CAAX which was generated by PCR with primers

osm960 (5′- GAAGCTTCGTACGCTGCAGG-3′) and osm207 (5′- cagcccgggTTATGAAATG ATGCATTTGTATAGTTCATCCATGCC-3′) from the pFA6a-GFP-kan ([79]; pSM674) plasmid. The correct plasmid was confirmed by sequencing. The sfGFP-RitC fragment encoding the sfGFP [82] fused to the RitC amphipathic helix (CPFFETSAAYRYYIDDVFHALVREIRR KEKEAVLAMEKKSKPKNSVWKRLKSPFRKKKDSVT) was amplified from pAV0605 vector using primers osm6872 (5′- ttcggatccCCGGGTTAATTAACTCCAAGGGTGAAGAGCTAT TTACT-3′) and osm6873 (5′-gaaatgatgcatATCAAGTTACACTGTCCTTTTTTTTTCGG-3′) and cloned into pSM1779 using BamHI and NsiI cloning sites to obtain plasmid pSM2528, which was used to express the Scd2-sfGFP-RitC from the weak p$^{nmt81}$ promoter.

In primer sequences, restriction sites are underlined, mutagenized sites are bold, and stop codon is bold italic. Plasmid maps are available upon request.

## Genetic analysis for synthetic lethality

Genetic interactions shown in Table 1 were assessed by tetrad dissection. Strains carrying *ras1* and *scd2* deletion were transformed with plasmids *pREP41-ras1* (pSM1143) and *pREP41-scd2* (pSM1351) before crosses to suppress sterility. Synthetic lethality was determined through statistical analysis as shown in S1 Table.

## Cell length and width measurements

For cell length and width measurements, cells were grown at 25 ˚C in EMM supplemented with amino acids as required. Exponentially growing cells were stained with calcofluor to visualize the cell wall and imaged on a DeltaVision platform described previously [83] or on a Leica epifluorescence microscope with 60x magnification. Measurements were made with ImageJ on septating cells. For each experiment, strains with identical auxotrophies were used. For cell length and width measurements shown in S2B Fig, cells were grown at 30 ˚C in 10 ml EMM in dark conditions.

## Microscopy

Images in Figs 3A, 3B, 9A (left), and 9C were acquired by using a DeltaVision platform (Applied Precision), previously described [70]. All other fluorescence microscopy experiments were done in a spinning disk confocal microscope, essentially as described [70, 83]. Image acquisition was performed on a Leica DMI6000SD inverted microscope equipped with an HCX PL APO 100X/1.46 numerical aperture oil objective and a PerkinElmer Confocal system. This system uses a Yokagawa CSU22 real-time confocal scanning head, solid-state laser lines, and a cooled 14-bit frame transfer EMCCD C9100-50 camera (Hamamatsu) and is run by Volocity (PerkinElmer). When imaging strains expressing the Opto$^{Q61L}$ and/or Opto systems, an additional long-pass color filter (550 nm, Thorlabs, Newton, NJ, USA) was used for brightfield (BF) image acquisition, in order to avoid precocious photoactivation caused by the white light.

Spinning disk confocal microscopy experiments (shown in Figs 2–4, 5D and 5E) were carried out using cell mixtures. Cell mixtures were composed by 1 strain of interest (the sample optogenetic strain, expressing or not an additional GFP-tagged protein) and 2 control strains (S3 Fig), namely:

1. RFP control: An RFP bleaching correction strain, expressing cytosolic CRY2PHR-mCherry.

2. GFP control: A wt strain expressing the same GFP-tagged protein as the strain of interest but without the optogenetic system. This strain was used both as negative control for cell-

side relocalization experiments and as GFP bleaching correction strain (in Figs 3–5 and S4–S7 Figs).

Strains were handled in dark conditions throughout. Red LED light was used in the room in order to manipulate strains and to prepare the agarose pads. Strains were cultured separately. Exponentially growing cells (O.D.$_{600nm}$ = 0.4–0.6) were mixed with 2:1:1 (strain of interest, RFP control, and GFP control) ratio and harvested by soft centrifugation (2 min at 1,600 rpm). The cell mixture slurry (1 μL) was placed on a 2% EMM-ALU agarose pad, covered with a #1.5-thick coverslip, and sealed with vaseline, lanolin, and paraffin (VALAP). Samples were imaged after 5–10 min of rest in dark conditions.

To assess the wavelength specificity for photoactivation of the Opto system (S2A Fig) in YSM3565 strains, cells were stimulated with blue (λ = 440 nm, λ = 488 nm) and green (λ = 561 nm) lasers. Samples were initially imaged for 20 s at 1-s intervals only in the RFP channel (λ = 561 nm). Laser stimulation was then performed using the microscope FRAP module (λ = 440 nm, λ = 488 nm, λ = 561 nm and no laser control). Cells were then monitored for another 40 s (1-s intervals) in the RFP channel. At the end of the time lapse, BF, GFP, and UV channel images were acquired.

The plasma-membrane recruitment dynamics of Opto$^{Q61L}$ and Opto systems were assessed using cell mixtures (S3 Fig). Protein recruitment dynamics was assessed by applying the 3 different photoactivating cycles listed below. Lasers were set to 100%; shutters were set to maximum speed; and in all instances, the RFP channel was imaged first, before the GFP channel. The duration of the experiment was equal regardless of the exposure time settings (≈15 s):

1. 50 ms: RFP channel (200 ms), GFP channel (50 ms). This constitutes 1 cycle (≈0.5 s). Thirty time points were acquired (≈0.5 s × 30 = 15.1 s).

2. 250 ms: RFP channel (200 ms), GFP channel (250 ms). This constitutes 1 cycle (≈0.7 s). Twenty-two time points were acquired (≈0.7 s × 22 = 15.1 s).

3. 500 ms: RFP channel (200 ms), GFP channel (500 ms). This constitutes 1 cycle (≈0.9 s). Seventeen time points were acquired (0.9 s × 17 = 15.5 s).

Endogenous GFP-tagged protein relocalization experiments were carried out using cell mixtures (control GFP was added, S3 Fig). Lasers were set to 100%; shutters were set to sample protection; and in all instances, the RFP channel was imaged first and then the GFP channel. RFP exposure time was always set to 200 ms, whereas the GFP exposure time varied depending on the monitored protein. Cells were monitored in these conditions for 90 s.

Spinning disk confocal time (sum) projections of 5 consecutive single-plane images are shown in Figs 1, 5C, 5F, 6, 7, 8 and 9 and S8 and S9 Figs. Single time point and max projection images are shown in Figs 2–4.

## Image analysis

All image-processing analyses were performed with ImageJ software (http://rsb.info.nih.gov/ij/). Image and time-lapse recordings were imported to the software using the Bio-Formats plugin (http://loci.wisc.edu/software/bio-formats). Time-lapse recordings were aligned using the StackReg plugin (http://bigwww.epfl.ch/thevenaz/stackreg/) according to the rigid-body method. All optogenetic data analyses were performed using MATLAB (R2018a), with scripts developed in-house.

Kymographs shown in Fig 3B and S2A Fig were generated with the MultipleKymograph (https://www.embl.de/eamnet/html/body_kymograph.html) ImageJ plugin. In Fig 3B, a 2-pixel-wide ROI was drawn along the cortex at the cell side. In S2A Fig, a 12-pixel-wide

($\approx$1 $\mu$m; 1 pixel = 0.083 $\mu$m) ROI was drawn crossing perpendicularly to the long axis of the cell. Fluorescence was averaged to 1-pixel-wide lines to construct the kymographs (parameter Linewidth = 1).

**Opto$^{Q61L}$ and Opto quantifications.** The plasma-membrane recruitment dynamics of Opto$^{Q61L}$ and Opto systems was assessed by recording the fluorescence intensity over an ROI that was 15 pixels long by 36 pixels wide (roughly 1.25 $\mu$m by 3 $\mu$m), drawn perpendicular to the plasma membrane of sample cells, from outside of the cell towards the cytosol (S3A Fig). The fluorescence intensity values across the length of the ROI were recorded over time in the RFP channel, in which each pixel represents the average of the width (36 pixels) of the ROI (3 replicates, 30 cells per replicate). Average background signal was measured from tag-free wt cells incorporated into the cell mixture (S3A Fig). The total fluorescence of the control RFP strain was also measured over time in order to correct for mCherry fluorophore bleaching. In both cases, the ROI encompassed whole cells, in which ROI boundaries coincide with the plasma membrane.

Photobleaching correction coefficient was calculated by the following formula:

$$RFP\ bleaching\ correction\ coefficient = {}^{(RFP\ Intensity_{tn} - NoGFPBckg_{tn})}\big/_{(RFP\ Intensity_{t0} - NoGFPBckg_{t0})} \quad (1)$$

where *RFP Intensity* is the signal measured from single RFP control cells, *NoGFPBckg* is the average background signal measured from tag-free cells, $t_n$ represents a given time point along the time course of the experiment, and $t_0$ represents the initial time point ($n = 30$ time points). These coefficients were corrected by a moving average smoothing method (moving averaged values = 5). RFP bleaching correction coefficient values calculated for individual RFP control cells were averaged in order to correct for bleaching of the RFP signal.

The fluorescence intensity values of optogenetic cells were corrected at each time point with the following formula:

$$\begin{aligned} RFP\ intensity \\ = ((Raw\ RFP\ signal_{tn} - NoGFPBckg_{tn})/RFP\ bleaching\ correction\ coefficient_{\ tn}) \end{aligned} \quad (2)$$

where *Raw RFP signal* is for the RFP values measured from sample strains, *NoGFPBckg* is the average background signal measured from tag-free cells, and $t_n$ represents a given time point along the time course of the experiment ($n = 30$ time points). The profiles resulting from these analyses are shown in Fig 2D, where the peaks of these profiles correspond to the plasma membrane. In order to get the net plasma-membrane recruitment profiles (Fig 2E), the fluorescence intensities from the peak $\pm$ 1 pixel were averaged and plotted over time.

$$Peak\ RFP\ intensity_{tn} = {}^{(RFP\ intensity_{peak-1pixel\ tn} + RFP\ intensity_{peak\ tn} + RFP\ intensity_{peak+1pixel\ tn})}\big/_{3} \quad (3)$$

$$Net\ P.M.recruitment\ Profile = (Peak\ RFP\ intensity_{tn} - Peak\ RFP\ intensity_{t0}) \quad (4)$$

Finally, the single-cell plasma-membrane recruitment half-times were calculated by fitting the normalized recruitment profiles with to the following formula:

$$RFP\ intensity(y) = a * (1 - e^{(-b*t)}) \quad (5)$$

$$Recruitment\ t_{1/2} = {}^{ln(0.5)}\big/_{b} \quad (6)$$

**Quantifications of the relocalization of GFP-tagged proteins to cell sides.** Endogenous GFP-tagged protein relocalization was assessed upon photoactivation of Opto$^{Q61L}$ and Opto

systems by recording the fluorescence intensity over an ROI that was 3 pixels wide by 36 pixels long ($\approx$0.25 μm by 3 μm) and drawn parallel to the cell-side cortex of sample cells (S3B Fig). The average fluorescence intensity values of both GFP and RFP channels were recorded over time from sample strains. In these particular experiments, a GFP control strain was included. These strains serve 2 purposes:

1. Calculation of the GFP bleaching correction coefficient (see below)

2. Negative control of the experiment; these strains carry the same endogenous GFP-tagged protein as the sample strain of the experiment, though lacking the optogenetic system. This controlled that GFP fluorescence changes were due to the optogenetic system and not caused by imaging per se. Control GFP strains were imaged in the same pad and analyzed in the same way as optogenetic cells (S3B Fig).

To derive photobleaching correction coefficients, the average camera background signals (*Bckg*) from 5 cell-free regions were measured as above, and fluorophore bleaching from RFP control and GFP control strains was measured at the cell side of control RFP and control GFP strains, for RFP and GFP channels, respectively.

$$\textbf{\textit{RFP bleaching correction coefficient}} = {}^{(RFP\ Intensity_{tn} - Bckg_{tn})} \big/ {}_{(RFP\ Intensity_{t0} - Bckg_{t0})} \qquad (7)$$

$$\textbf{\textit{GFP bleaching correction coefficient}} = {}^{(GFP\ Intensity_{tn} - Bckg_{tn})} \big/ {}_{(GFP\ Intensity_{t0} - Bckg_{t0})} \qquad (8)$$

where *RFP Intensity* and *GFP Intensity* stand for the signal measured from RFP control and GFP control cells, respectively; $t_n$ represents a given time point along the time course of the experiment; and $t_0$ represents the initial time point ($n = 30$ time points). These coefficients were corrected by a moving average smoothing method, as above.

The fluorescence intensity values of optogenetic cells in both GFP and RFP channels were independently analyzed as follows. First, GFP and RFP signals were background and bleaching corrected, using Eqs 7 and 8 for the RFP and GFP channels, respectively:

$$\textbf{\textit{P.M.GFP/RFP value}}_{tn} = ((Raw\ signal_{tn} - Bckg_{tn})/bleaching\ correction\ coefficient\ _{tn}) \qquad (9)$$

where *Raw signal* intensity represents the GFP or RFP raw values at the cell-side cortex, *Bckg* stands for the average fluorescence intensity of 5 independent cell-free regions, and $t_n$ represents a given time point along the time course of the experiment ($n = 30$ time points). The net fluorescence intensity at the cell-side cortex was then calculated for both GFP and RFP signals.

$$\textbf{\textit{Net P.M.GFP/RFP value}}_{tn} = (Fluorescence\ intensity_{tn} - Fluorescence\ intensity_{t0}) \qquad (10)$$

From here on, RFP and GFP signals were treated differently. Single-cell plasma-membrane RFP profiles from Eq 10 were individually normalized and fitted to Eq 5 in order to extrapolate the parameter b. Using Eq 6, recruitment half-times of Opto and Opto$^{Q61L}$ systems were calculated. Because of lower signal-to-noise ratio of the weak GFP fluorescence, plasma-membrane GFP profiles from Eq 10 were averaged ($n > 20$ profiles per experiment), and the initial 45 s of the average profile was used to extract the half-time of plasma-membrane relocalization of endogenous GFP-tagged proteins using Eqs 5 and 6. Three experimental replicates were performed and are plotted on Fig 3D.

**Quantifications of the relocalization of GFP-tagged proteins from cell tips.** Scd1-3GFP tip signal analyses (Fig 4B) were performed from the same time-lapse recordings as cell-side relocalization experiments. Scd1-3GFP tip signal was recorded over an ROI that was 3 pixels wide by 6–12 pixels long ($\approx$0.25 μm by 0.5–1 μm) and drawn at the tip of the cells. To derive

photobleaching correction coefficients, the average camera background signals (*Bckg*) from 5 cell-free regions were measured as before, and GFP bleaching from GFP control strain was measured at the cell tip.

$$\textbf{\textit{Tip GFP bleaching correction coefficient}} = {}^{(GFP\ Intensity_{tn} - Bckg_{tn})} \big/ {}_{(GFP\ Intensity_{t0} - Bckg_{t0})} \qquad (11)$$

where *GFP Intensity* stands for the signal measured from the tip of GFP control cells, $t_n$ represents a given time point along the time course of the experiment, and $t_0$ represents the initial time point ($n = 30$ time points). This coefficient was corrected by a moving average smoothing method, as before.

The tip GFP fluorescence intensity values of optogenetic cells were analyzed as follows. First, GFP signals were background and bleaching corrected, using Eq 12:

$$\textbf{\textit{Tip GFP value}}_{tn} = (\textit{Tip Raw signal}_{tn} - \textit{Bckg}_{tn}) / \textit{Tip GFP bleaching correction coefficient }_{tn} \quad (12)$$

where *Tip Raw signal* intensity represents the GFP raw values at the cell tip, *Bckg* stands for the average fluorescence intensity of 5 independent cell-free regions, and $t_n$ represents a given time point along the time course of the experiment ($n = 30$ time points). The tip fluorescence intensities of single optogenetic strains were then normalized relative to their GFP values at the initial time point.

$$\textbf{\textit{Normalized tip GFP value}}_{tn} = (\textit{Tip GFP value}_{tn} / \textit{tip GFP}_{t0}) \qquad (13)$$

Eventually, average Scd1-3GFP tip signal was calculated (>15 cells, Fig 4B).

Cdc42-sfGFP[SW] tip signal analyses (Fig 3G) were performed from the same time-lapse recordings as cell-side relocalization experiments. Cdc42-sfGFP[SW] tip signal was recorded over an ROI that was 3 pixels wide by 6–12 pixels long ($\approx$0.25 μm by 0.5–1 μm) and drawn at the tip of the cells at 0 s, 300 s, 600 s, and 900 s. The data were analyzed as described for Scd1-3GFP tip signal, with the exception that only 4 time points were considered instead of 30 time points.

Quantification of cortical fluorescence at the cell ends in Figs 1 and 7 was done by using the sum projection of 5 consecutive images. The intensity of a 3-pixel-wide segmented line along the cell tip was collected and corrected for camera noise background. The profiles were aligned to the geometrical center of the cell tip. Quantifications in Fig 7C and 7D are the average value of 5 pixels around the geometrical center.

In Figs 3H and 4C, the initial time point of the time lapse was used to subtract the fluorescence signal along the subsequent time points using image calculator (ImageJ). The time lapse was previously corrected for photobleaching by the bleach corrector plugin using the exponential fitting method [84]. The displayed images were pseudocolored using the "gem" option (imageJ).

Figures were assembled with Adobe Photoshop CS5 and Adobe Illustrator CS5. All error bars on bar graphs are standard deviations. For statistical analysis, in Figs 3–5, cumulative GFP signal (addition of GFP signal along the 30 time points of the time lapse) was calculated from single-cell traces of Opto, Opto[Q61L], and GFP control cells. For statistical analysis, single-cell cumulative GFP signals of the entire dataset (3 independent experiments combined) were considered, without averaging. Data normality was assessed by the Lilliesfors test and significance by pairwise Kruskal-Wallis analysis. *p*-Values show significance of differences between Opto[Q61L] and Opto cells, unless indicated otherwise. In S2B Fig, data normality was assessed by the Lilliesfors test and significance by pairwise Kruskal-Wallis analysis. *t* Test was used in Figs 1, 6, 8 and 9 and S8 Fig. For aspect-ratio measurements, box plots show the ratio between cell length and cell width, and bar graph error bars show standard deviation. On each box, the

central red mark indicates the median; the bottom and the top edges indicate the 25th and 75th percentiles, respectively; and the whiskers extend to the most extreme data points, not considering outliers, which are plotted individually using the red "+" symbol. All experiments were done at least 3 independent times.

## Supporting information

**S1 Fig. Mutant cells lacking Scd2 and Gef1 are viable (related to Fig 1).** (A-C) Tenfold serial dilutions of strains with indicated genotypes spotted on YE-containing plates incubated at the specified temperatures. YE, yeast extract.
(PDF)

**S2 Fig. Implementing the CRY2PHR-CIBN optogenetic system in *S. pombe* cells (related to Fig 2).** (A) Blue light–dependent cortical recruitment of CRY2PHR-mCherry in cells expressing CIBN-mTagBFP2 targeted to the plasma membrane. Scale bar = 5 µm. The scheme on the right explains the experimental design demonstrating blue-light specificity. (B) Cell length and width measurements, aspect ratio, and bipolarity of calcofluor-stained cells grown in the dark. The CRY2PHR-CIBN optogenetic system does not cause changes in cell dimensions. Cytosolic Cdc42$^{Q61L}$ causes moderate cell length shortening, with significant impact on the cell aspect ratio, irrespective of the presence of CIBN ($p^{Cdc42Q61L} = 0.02$; $p^{OptoQ61L} = 0.003$ relative to wild-type cells; other comparisons yield $p^{WTvsCIBN} = 0.3$; $p^{WTvsCRYPHR-mCh} = 0.1$; $p^{WTvsOpto} = 0.1$. Monopolar and bipolar growth were assessed on septated cells. All underlying numerical values are available in S10 Data.
(PDF)

**S3 Fig. Cell mixtures for data analysis.** (A) Representative initial (t = 0 s) merged image of plasma-membrane recruitment dynamic experiments performed for Opto$^{Q61L}$ and Opto systems. Shown is the Opto system prior to stimulation with 50-ms GFP laser pulses. Cells labeled as 1 are the tag-free cells used to correct the raw data for *NoGFPBckg* (see Eqs 1 and 2 in Methods). Cells labeled as 2 are the RFP control cells used to calculate the RFP bleaching coefficient (see Eq 1 in Methods). Cells labeled as 3 are the optogenetic cells from which plasma-membrane recruitment dynamics were measured (*Raw RFP signal* parameter in Eq 2 in Methods). ROI = 15 pixels long by 36 pixels wide (roughly 1.25 µm by 3 µm). (B) (Top) Representative initial (t = 0 s) merged image of the relocalization of GFP-tagged proteins to cell sides experiments. Shown are wild-type and Opto CRIB-3GFP cells prior stimulation with blue light. ROIs labeled as 1 show the cell-free regions used to correct the raw data for *Bckg* (see Eqs 7, 8, and 9 in Methods). Cells labeled as 2 are RFP control cells used to calculate RFP bleaching coefficient (see Eq 7 in Methods). Cells labeled as 3–4 are GFP control cells used to calculate GFP bleaching coefficient and as control cells for cell-side relocalization of GFP-tagged endogenous proteins (see Eq 8 in Methods). Cells labeled as 5 are optogenetic cells from which cell-side relocalization of GFP-tagged endogenous proteins was monitored (see Eqs 9–13 in Methods). ROI = 3 pixels wide by 36 pixels long (≈0.25 µm by 3 µm). (Bottom) GFP channel from the merged image shown above to illustrate the background fluorescence signal in non-GFP-containing cells (labeled as 2 and 6). Bars = 10 µm. ROI, region of interest.
(PDF)

**S4 Fig. Controls and single-cell traces for optogenetic protein recruitment in *scd2+* cells (related to Fig 3).** (A) Single-cell traces corresponding to the average plots shown in Fig 3C. The left column shows the average RFP signal at the plasma membrane of wild-type Opto$^{Q61L}$ and Opto cells. The 3 other graphs show, from left to right, single-cell GFP traces of Opto$^{Q61L}$, Opto, and wild-type control cells for CRIB-3GFP, Pak1-sfGFP, Scd2-GFP, and Scd1-3GFP in

otherwise wild-type cells. (B) Single-cell traces corresponding to the average plots shown in Fig 3E. The left column shows the average RFP signal at the plasma membrane of Opto$^{WT}$ (top) and Opto$^{T17N}$ (bottom) cells. The 3 other graphs show, from left to right, single-cell GFP traces of Opto$^{WT}$ (top) and Opto$^{T17N}$ (bottom), Opto, and wild-type control cells for endogenous Scd2-GFP. Note that the Opto$^{WT}$ and Opto$^{Q61L}$ experiments were performed in parallel, and thus, the Opto and wild-type control single-cell GFP traces are identical to those shown for Scd2-GFP in (A). (C) Single-cell traces corresponding to the average plots shown in Fig 3F–3G. The left column shows the average RFP signal at the plasma membrane of Opto$^{Q61L}$ and Opto cells. The 3 other graphs show, from left to right, single-cell GFP traces of Opto$^{Q61L}$, Opto, and wild-type control cells for endogenous Cdc42-sfGFP$^{SW}$. $N$ = 3 experiments with $n > 20$ cells. All underlying numerical values are available in S11 Data. (PDF)

**S5 Fig. Scd2 and Pak1 are recruited by Opto$^{Q61L}$ in absence of GEFs (related to Fig 3).** (A) Opto$^{Q61L}$-induced cell-side accumulation of Scd2-GFP in *scd1Δ* (top) and *gef1Δ* (bottom) cells. $N$ = 3; $n > 20$ cells per experiment; $p^{scd1Δ} = 2e^{-24}$ and $p^{gef1Δ} = 7.6e^{-16}$. (B) Single-cell traces corresponding to the average plots shown in (A). The left column shows the average RFP signal at the plasma membrane in Opto$^{Q61L}$ and Opto cells of the indicated genotype. The 3 other graphs show, from left to right, single-cell GFP traces of Opto$^{Q61L}$, Opto, and control cells for Scd2-GFP in *scd1Δ* and *gef1Δ* cells. $N$ = 3 experiments; $n > 20$ cells. (C) Opto$^{Q61L}$-induced cell-side accumulation of Pak1-GFP in *scd1Δ*. $N$ = 3; $n > 20$ cells per experiment; $p = 1.3e^{-22}$. (D) Single-cell traces corresponding to the average plots shown in (C). The left column shows the average RFP signal at the plasma membrane in Opto$^{Q61L}$ and Opto cells of indicated genotype. The 3 other graphs show, from left to right, single-cell GFP traces of Opto$^{Q61L}$, Opto, and control cells for Pak1-sfGFP in *scd1Δ* cells. $N$ = 3 experiments; $n > 20$ cells. All underlying numerical values are available in S12 Data. (PDF)

**S6 Fig. Controls and single-cell traces for optogenetic protein recruitment in *scd2Δ* cells (related to Fig 4).** (A) Single-cell traces corresponding to the average plots shown in Fig 4A. The left column shows the average RFP signal at the plasma membrane of *scd2Δ* Opto$^{Q61L}$ and Opto cells. The 3 other graphs show, from left to right, single-cell GFP traces of Opto$^{Q61L}$, Opto, and control cells for CRIB-3GFP, Pak1-sfGFP, and Scd1-3GFP in *scd2Δ* cells. (B) Single-cell traces corresponding to the average plots shown in Fig 4D. The left column shows the average RFP signal at the plasma membrane of *scd2Δ* Opto$^{Q61L}$ and Opto cells. The 3 other graphs show, from left to right, single-cell GFP traces of *scd2Δ* Opto$^{Q61L}$, Opto, and wild-type control cells for endogenous Cdc42-sfGFP$^{SW}$. $N$ = 3 experiments with $n > 20$ cells. All underlying numerical values are available in S13 Data. (PDF)

**S7 Fig. Controls and single-cell traces for optogenetic protein recruitment in *scd2* mutant allele cells (related to Fig 5).** (A) Single-cell traces corresponding to the average plots shown in Fig 5D. The left column shows the average RFP signal at the plasma membrane of *scd2$^{1-266}$*, *scd2$^{275-536}$*, and *scd2$^{k463A}$* Opto$^{Q61L}$ and Opto cells. The 3 other graphs show, from left to right, single-cell GFP traces of Opto$^{Q61L}$, Opto, and control cells for Scd2$^{1-266}$-eGFP, Scd2$^{275-536}$-eGFP, and Scd2$^{K463A}$-eGFP. (B) Single-cell traces corresponding to the average plots shown in Fig 5E. The left column shows the average RFP signal at the plasma membrane of *scd2$^{275-536}$* and *scd2$^{k463A}$* Opto$^{Q61L}$ and Opto cells. The 3 other graphs show, from left to right, single-cell GFP traces of Opto$^{Q61L}$, Opto, and control cells for Scd1-3GFP in *scd2$^{275-536}$* and *scd2$^{K463A}$* cells. $N$ = 3 experiments with $n > 20$ cells. All

underlying numerical values are available in S14 Data.
(PDF)

**S8 Fig. Expression of Scd1-Pak1 bridge suppresses the lethality of *scd2Δ ras1Δ gef1Δ* mutants (related to Fig 6).** (A) Tenfold serial dilutions of strains with indicated genotypes spotted on YE-containing plates incubated at the specified temperatures. (B) Mean cell length and width at division (left), and aspect ratio (right), of strains with indicated genotypes. $N = 3$ experiments with $n > 30$ cells; ***$3.5e^{-48} \leq p \leq 2e^{-7}$. Bar graph error bars show standard deviation; box plots indicate the median, 25th and 75th percentiles, and most extreme data points, not considering outliers, which are plotted individually using the red "+" symbol. All underlying numerical values are available in S15 Data. YE, yeast extract.
(PDF)

**S9 Fig. Cdc42 activity is reduced in the absence of Ras1 (related to Fig 7).** (A) Localization of Scd1-3GFP (B/W inverted images) and CRIB-3mCherry (magenta) in *gef1Δ, scd2Δ, scd2Δ, gef1Δ, ras1Δ, gef1Δ,* and *scd2Δ, ras1Δ* cells expressing the *scd1-pak1 bridge*. (B) Cortical tip profiles of CRIB-3mCherry (left) and Scd1-3GFP (middle) and ratio of Scd1-3GFP and CRIB-3mCherry (right) fluorescence at the cell tip of strains as in (A); $n = 30$ cells. Bar = 2 μm. All underlying numerical values are available in S16 Data. B/W, black and white.
(PDF)

**S1 Movie. Related to Fig 1F: Gef1 forms unstable zones of Cdc42 activity in the absence of Scd2 and Ras1.** Localization of CRIB-GFP (green, left) and Gef1-tdTomato (magenta, middle) and colocalization of CRIB-GFP and Gef1-tdTomato (merged image, right) in *ras1Δ, scd2Δ* double-mutant cells. Scale bar = 2 μm.
(AVI)

**S2 Movie. Related to Fig 2G: Opto^Q61L induces isotropic growth under constant blue-light activation.** Opto^Q61L and Opto^WT (blue and green cells respectively, in the left panel showing merge bright-field, GFP, and UV channels) cells growing under periodic (every 10 min) blue-light photostimulation. The left panel shows the cortical recruitment of the Cdc42-mCherry-CRY2PHR moiety (magenta, right). Scale bar = 2 μm.
(MOV)

**S3 Movie. Related to Fig 3C: Opto^Q61L induces cell-side relocalization of Cdc42-GTP sensor CRIB-3GFP, Cdc42 effector Pak1-sfGFP, scaffold protein Scd2-GFP, and Cdc42 GEF Scd1-3GFP.** Cell-side relocalization of CRIB-3GFP, Pak1-sfGFP, Scd2-GFP, and Scd1-3GFP (inverted B/W images, from left to right) in Opto^Q61L (magenta, bottom) cells upon blue-light activation. Opto cells (magenta, top) are shown as control. Scale bar = 2 μm. B/W, black and white.
(MOV)

**S4 Movie. Related to Fig 4A: Scd2 scaffold is essential to recruit Cdc42 GEF Scd1 to active Cdc42 sites.** Cell-side relocalization of CRIB-GFP and Pak1-sfGFP but not Scd1-3GFP (inverted B/W images, from left to right) in Opto^Q61L *scd2Δ* (magenta, bottom) cells upon blue-light activation. Opto *scd2Δ* cells (magenta, top) are shown as control. Scale bar = 2 μm. B/W, black and white.
(MOV)

**S5 Movie. Related to Fig 6B–6D: *scd2Δ ras1Δ gef1Δ* cells expressing the Scd1-Pak1 bridge growth in a bipolar manner.** Localization of Scd1-3GFP (green, left), localization of Pak1-GBP-mCherry (magenta, middle), and colocalization of Scd1-3GFP and Pak1-GBP-mCherry (merged image, right) in *scd2Δ ras1Δ gef1Δ* triple-mutant cells. Scale bar = 2 μm. GBP,

GFP-binding protein.
(AVI)

**S1 Table. Genetic interaction of *scd1Δ*, *scd2Δ*, *ras1Δ*, and *gef1Δ*.** Supporting information for Table 1, showing the genetic crosses performed and the number of spores of indicated genotypes analyzed by tetrad dissection.
(XLSX)

**S2 Table. List of strains used in this study.**
(XLSX)

**S1 Data. Values for each data point to create the graphs in Fig 1.**
(ZIP)

**S2 Data. Values for each data point to create the graphs in Fig 2.**
(ZIP)

**S3 Data. Values for each data point to create the graphs in Fig 3.**
(ZIP)

**S4 Data. Values for each data point to create the graphs in Fig 4.**
(ZIP)

**S5 Data. Values for each data point to create the graphs in Fig 5.**
(ZIP)

**S6 Data. Values for each data point to create the graphs in Fig 6.**
(ZIP)

**S7 Data. Values for each data point to create the graphs in Fig 7.**
(ZIP)

**S8 Data. Values for each data point to create the graphs in Fig 8.**
(ZIP)

**S9 Data. Values for each data point to create the graphs in Fig 9.**
(ZIP)

**S10 Data. Values for each data point to create the graphs in S2 Fig.**
(ZIP)

**S11 Data. Values for each data point to create the graphs in S4 Fig.**
(ZIP)

**S12 Data. Values for each data point to create the graphs in S5 Fig.**
(ZIP)

**S13 Data. Values for each data point to create the graphs in S6 Fig.**
(ZIP)

**S14 Data. Values for each data point to create the graphs in S7 Fig.**
(ZIP)

**S15 Data. Values for each data point to create the graphs in S8 Fig.**
(ZIP)

**S16 Data. Values for each data point to create the graphs in S9 Fig.**
(ZIP)

## Acknowledgments

We thank Ken Sawin (University of Edinburgh) and Chandra Tucker (University of Colorado) for plasmids, Serge Pelet (University of Lausanne) for help with MatLab scripts, and Serge Pelet and Veneta Gerganova for careful reading of the manuscript.

## Author contributions

AV conceived and initiated the optogenetic modules and showed the first proof of concept. IL performed all optogenetic experiments (Figs 2–4, 5D, 5E and S2–S7 Figs), with initial help and guidance from AV, and all related quantifications in MatLab. LM performed all other experiments, except for Figs 6E and 6F and 9C and 9D, which were done by AV. VV helped with strain construction and tetrad dissection. SGM provided supervision, acquired funding, and wrote the manuscript with help of LM, IL, and AV.

## Author Contributions

**Conceptualization:** Laura Merlini, Aleksandar Vještica, Sophie G. Martin.

**Funding acquisition:** Sophie G. Martin.

**Investigation:** Iker Lamas, Laura Merlini, Aleksandar Vještica, Vincent Vincenzetti.

**Methodology:** Iker Lamas, Laura Merlini, Aleksandar Vještica.

**Project administration:** Sophie G. Martin.

**Supervision:** Aleksandar Vještica, Sophie G. Martin.

**Writing – original draft:** Laura Merlini, Sophie G. Martin.

**Writing – review & editing:** Iker Lamas, Laura Merlini, Aleksandar Vještica, Sophie G. Martin.

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
