## [Editor Report · Decision Letter 0]

29 Jul 2019

Dear Sophie, 

Thank you for submitting your manuscript entitled "Optogenetics reveals Cdc42 local activation by scaffold-mediated positive feedback and Ras GTPase" for consideration as a Research Article by PLOS Biology.

Your manuscript has now been evaluated by the PLOS Biology editorial staff as well as by an Academic Editor with relevant expertise and I am writing to let you know that we would like to send your submission out for external peer review.

*Please be aware that, due to the voluntary nature of our reviewers and academic editors, manuscript review may be subject to delays during this busy summer travel season. Thank you for your patience.*

**Important**: Please also see below for further information regarding completing the MDAR reporting checklist. The checklist can be accessed here: https://plos.io/MDARChecklist

Please re-submit your manuscript and the checklist, within two working days, i.e. by Jul 31 2019 11:59PM.

Kind regards,

Hashi Wijayatilake, PhD,

Managing Editor

PLOS Biology

INFORMATION REGARDING THE REPORTING CHECKLIST:

PLOS Biology is pleased to support the "minimum reporting standards in the life sciences" initiative (https://osf.io/preprints/metaarxiv/9sm4x/). This effort brings together a number of leading journals and reproducibility experts to develop minimum expectations for reporting information about Materials (including data and code), Design, Analysis and Reporting (MDAR) in published papers. We believe broad alignment on these standards will be to the benefit of authors, reviewers, journals and the wider research community and will help drive better practise in publishing reproducible research. 

We are therefore participating in a community pilot involving a small number of life science journals to test the MDAR checklist. The checklist is intended to help authors, reviewers and editors adopt and implement the minimum reporting framework. 

IMPORTANT: We have chosen your manuscript to participate in this trial. The relevant documents can be located here:

MDAR reporting checklist (to be filled in by you): https://plos.io/MDARChecklist

**We strongly encourage you to complete the MDAR reporting checklist and return it to us with your full submission, as described above. We would also be very grateful if you could complete this author survey:

https://forms.gle/seEgCrDtM6GLKFGQA

Additional background information:

Interpreting the MDAR Framework: https://plos.io/MDARFramework

Please note that your completed checklist and survey will be shared with the minimum reporting standards working group. However, the working group will not be provided with access to the manuscript or any other confidential information including author identities, manuscript titles or abstracts. Feedback from this process will be used to consider next steps, which might include revisions to the content of the checklist. Data and materials from this initial trial will be publicly shared in September 2019. Data will only be provided in aggregate form and will not be parsed by individual article or by journal, so as to respect the confidentiality of responses. 

Please treat the checklist and elaboration as confidential as public release is planned for September 2019.

We would be grateful for any feedback you may have.

---

## [Decision Letter · Decision Letter 1]

23 Aug 2019

Dear Sophie,

Thank you very much for submitting your manuscript "Optogenetics reveals Cdc42 local activation by scaffold-mediated positive feedback and Ras GTPase" for consideration as a Research Article at PLOS Biology. Your manuscript has been evaluated by the PLOS Biology editors, an Academic Editor with relevant expertise, and by several independent reviewers.

The reviewers are all generally positive, especially about this optogenetic approach in fission yeast. They do however have some requests to help better support the current conclusions. In light of the reviews, we will not be able to accept the current version of the manuscript, but we would welcome resubmission of a revised version that takes into account the reviewers' comments. 

Please note that we cannot make any decision about publication until we have seen the revised manuscript and your response to the reviewers' comments. Your revised manuscript is also likely to be sent for further evaluation by the reviewers.

Your revisions should address the specific points made by each reviewer. Please submit a file detailing your responses to the editorial requests and a point-by-point response to all of the reviewers' comments that indicates the changes you have made to the manuscript. In addition to a clean copy of the manuscript, please upload a 'track-changes' version of your manuscript that specifies the edits made. This should be uploaded as a "Related" file type. You should also cite any additional relevant literature that has been published since the original submission and mention any additional citations in your response. 

Before you revise your manuscript, please review the following PLOS policy and formatting requirements checklist PDF: http://journals.plos.org/plosbiology/s/file?id=9411/plos-biology-formatting-checklist.pdf. It is helpful if you format your revision according to our requirements - should your paper subsequently be accepted, this will save time at the acceptance stage.

Please note that as a condition of publication PLOS' data policy (http://journals.plos.org/plosbiology/s/data-availability) requires that you make available all data used to draw the conclusions arrived at in your manuscript. If you have not already done so, you must include any data used in your manuscript either in appropriate repositories, within the body of the manuscript, or as supporting information (N.B. this includes any numerical values that were used to generate graphs, histograms etc.). For an example see here: http://www.plosbiology.org/article/info%3Adoi%2F10.1371%2Fjournal.pbio.1001908#s5.

For manuscripts submitted on or after 1st July 2019, we require the original, uncropped and minimally adjusted images supporting all blot and gel results reported in an article's figures or Supporting Information files. We will require these files before a manuscript can be accepted so please prepare them now, if you have not already uploaded them. Please carefully read our guidelines for how to prepare and upload this data: https://journals.plos.org/plosbiology/s/figures#loc-blot-and-gel-reporting-requirements.

Upon resubmission, the editors will assess your revision and if the editors and Academic Editor feel that the revised manuscript remains appropriate for the journal, we will send the manuscript for re-review. We aim to consult the same Academic Editor and reviewers for revised manuscripts but may consult others if needed.

We expect to receive your revised manuscript within two months. Please email us (plosbiology@plos.org) to discuss this if you have any questions or concerns, or would like to request an extension. At this stage, your manuscript remains formally under active consideration at our journal; please notify us by email if you do not wish to submit a revision and instead wish to pursue publication elsewhere, so that we may end consideration of the manuscript at PLOS Biology.

When you are ready to submit a revised version of your manuscript, please go to https://www.editorialmanager.com/pbiology/ and log in as an Author. Click the link labelled 'Submissions Needing Revision' where you will find your submission record. 

Sincerely,

Hashi Wijayatilake, PhD, 

Managing Editor

PLOS Biology

REVIEWS:

Reviewer #1: 

This excellent paper uses optogenetic approaches to dissect the polarity control circuit in fission yeast. The major conclusions are that Ras1 acts both to localize and stimulate the Cdc42 GEF Scd1, and that the scaffold protein Scd2 acts to promote positive feedback by linking GTP-Cdc42 effector Pak1 to Scd1. These conclusions closely parallel similar conclusions about the budding yeast system, but the paper nevertheless represents a major advance for two reasons. First, it was unclear how generally the proposed polarity circuit would apply, and given the evolutionary distance between the two yeasts this is an important extension. Second, the use of cutting-edge methods cements the model by presenting findings not previously shown. The paper is also notable for the care taken to provide appropriate controls, and to conduct rigorous image analysis. Overall, this is a thorough and well-executed study on an important problem. I had a number of questions and suggestions as I read through, with the most important being #3-4 which affect a major conclusion.

Questions and suggestions:

1. The Opto-42 looks patchy on the membrane in Fig. 2G as compared to Opto alone in Fig. 2B,C—any idea why? Is there a tendency to aggregate? Or a correlation (or anti-correlation) with other patchy structures like eisosomes or nodes? Or exclusion by cortical ER? 

2. The lateral recruitment of GFP-tagged polarity proteins documented in Fig. 3A and S3 appears quite weak. While the care taken with controls and image analysis and statistics makes a convincing case that there IS recruitment, I was left wondering why it was so weak relative to the tip signal, and how such weak recruitment could lead to the longer term depolarized growth in Fig. 1? 

3. The recruitment of Scd1 to cell sides by optoQ61L was especially weak, and seemed limited to a few puncta-much more so than CRIB or Scd2 signals. Do the authors have an explanation for this? Or for the internal blobs and filaments seen with this probe?

4. An important conclusion, that Scd2 mediates positive feedback recruitment of Scd1, was hard for me to assess. The main problems are that Scd1 recruitment was so weak even with Scd2 (see point 3) and that Scd1 signals are low in scd2 deletes, as the authors acknowledged. I’m not sure that the failure to see “competition” with tip signal gets around these issues. One potential way to make this more convincing would be to show whether Cdc42 gets recruited to cell sides by OptoQ61L in the absence of Scd2.

5. The localization of Scd2K463A and Scd21-266 are different—why? If I understood correctly, both should bind Cdc42 and neither should bind Scd1, so why the difference? Are they expressed at different levels? 

6. Fig. 6 exploits the Scd1-Pak1 bridge to bypass the role of Ras1 in Scd1 recruitment and show that Ras1 increases GTP-Cdc42 at cell poles in a manner not fully explained by the level of Scd1. The authors imply/conclude (e.g. section and Figure title, as well as Discussion) that Ras1 promotes activation of Scd1. But mechanistically, couldn’t this work by other means—like Ras1-mediated suppression of Cdc42 GAPs?

7. p. 12: The authors state that scd2 gap1 mutants localize Scd1 and CRIB “all around the membrane”, citing Fig. 7D,G. But that is not what it looks like to this reader: the signals in 7D are mainly found in patches and have very different concentrations around the membrane; and 7G is not about localization of probes.

8. Fig. 7F shows a very interesting finding that Scd1-Pak1 and Gef1-Pak1 bridge strategies yield very different outcomes in the gap1 scd2 context. They conclude that Gef1 is not involved in positive feedback, but I don’t think this experiment addresses that. Rather, it shows an artificial bridge that SHOULD yield positive feedback by localizing Gef1 activity to sites enriched for active Cdc42 doesn’t work. Why wouldn’t Gef1-Pak1 work? Is it able to rescue scd1 gef1 mutants? 

9. To actually address whether Gef1 is involved in positive feedback, it would be interesting to know whether Gef1 is recruited to cell sides by OptoQ61L.

10. Fig. 7H,I: This experiment introduces a new reagent Scd2-CAAX but does not validate that it really localizes all around the membrane: such validation is needed to interpret the experiment.

Issues with presentation:

1. P-values are provided in Figure Legends, but there was no description or justification of what statistical test was applied where.

2. It would be good to see images of Cdc42 recruitment (as in Fig. 3A) to back up the quantification provided in Fig. 3C. Unlike for the other probes, one would expect to see a basal level of Cdc42 at the membrane.

3. The authors have taken the excellent approach of including negative control cells mixed in with the cells of interest on the same imaging slab, which eliminates potential day-to-day and slab-to-slab differences as a basis for their observations. While this is explained in the methods, it would be beneficial for the average reader to include a mention in the Figure Legends or main text. This would alert the reader to the significance of the BFP-only and RFP-only cells in Fig. 2B, for example.

4. The use of inverted black on white images is excellent and allows the reader to optimally appreciate the results. And the signal from optogenetic system constructs is so strong that color panels are OK there, but why use the much less clear magenta-on-black for various panels in Fig. 5-7?

5. p. 34 top: Should “cell side tip” be “cell tip”?

--

Reviewer #2: 

This manuscript examines the mechanisms of Cdc42 activation in fission yeast S. pombe. The authors use both conventional genetics and optogenetics to dissect the Scd1/Scd2/Pak1 polarity complex. The optogenetic studies indicate that membrane recruitment of an activated form Cdc42 appears to be able to recruit components Scd1, Scd2, and Pak1 and that that Scd1 recruitment appears to require Scd2. The authors claim direct recruitment of Scd2 by Cdc42, however these results are not shown in full and what is shown is not fully compelling. Overall the results deepen the parallels between the behavior of these Cdc42 regulators in budding and fission yeast. The strongest part of the manuscript are the conventional cell biological and genetic studies that demonstrate that Ras1 plays a role in both recruitment and activation of Scd1.

The authors use the Cry2/Cib system to recruit an activated allele of Cdc42 lacking its CaaX box. The authors appear to imply that this recruited Cdc42 is normally active. However, this assumption is not well founded as this mode of tethering may not result in a Cdc42 molecule in its normal proximity, and perhaps orientation, relative to the plasma membrane. While the tagged molecule is robustly, uniformly, and rapidly recruited to the cortex, the recruitment of polarity proteins is far slower, far less robust, and highly patchy, suggesting that only a small proportion of recruited molecules are functional. There is no mention in the text of the low degree of spatial correlation between the recruited Cdc42 and its interacting partners. The recruited Cdc42 does have some biological activity as the cells exhibit light dependent depolarization. However, this could be due to either a gain of function effect (global Cdc42 activation) or a dominant negative effect (sequestering and inactivating polarity proteins) or a combination thereof; it is not clear which is correct. However, the authors state, "This transition from rod to round shape is a clear evidence of isotropic growth triggered by the recruitment of Cdc42 activity to the plasma membrane in a light-dependent manner." The temporal resolution of the movie provided (S2) (1 hour intervals) and the absence of a marker for the growing end limits the utility of this movie. I would not assess the evidence as “clear”.

In addition, it is not apparent that the polarity complexes that are recruited in a Cdc42-dependent manner are highly active as the amount of CRIB-GFP recruited to the cortex is similar in the presence and absence of Scd2 (compare Fig 3A vs 4A). A possible explanation is that the recruited active Cdc42 does not recruit a particularly active GEF.

The authors state that the recruited Cdc42 demonstrates positive feedback which had not been shown previously. Strictly speaking this claim is valid as some endogenous Cdc42 is recruited to cell sides (fig 4C), though only the quantification and not the raw imaging data are shown for this conclusion. Overall, It is difficult to make this claim in the absence of evidence showing proper light dependent polarization, as was shown with the Scd1/Pak1 bridged molecule.

In this vein, the authors state, that prior experiments in budding yeast did not "did not directly test whether Cdc42 promotes its own activation." Again, formally this is correct, but the aforementioned paper did demonstrate that the ability of the Cdc24 GEF to induce cell polarization required its GEF activity, but not the ability to bind Bem1, implying that active Cdc42 is required for positive feedback. Overall, the authors do not summarize the more salient results of this paper. They focus instead on the cell cycle control aspects which are not particularly relevant to the present paper and don't mention the direct demonstration of light directed polarization and positive feedback among polarity proteins or the winner take all behavior of the cell even in the presence of two distinct foci of active Cdc42, which are far more applicable to this manusript. 

The authors stated they were unable to use their optogenetic system to perform local recruitment assays which precludes its use to address many interesting questions related to the oscillation of Cdc42 activation and even to test whether local recruitment is sufficient to induce cell polarization. It is possible that illumination with excessive levels of light resulted in global activation. The high rates of diffusion in yeast cells does not preclude the generation of a local pool of activated Cdc42.

Related to figure 4, the recruitment of Scd2 1-266 is a key result, but again, the raw data not shown. The authors conclude, "This suggests Cdc42-GTP interaction is a major contributor to Scd2 localization." and "Thus, Scd2 is the scaffold that promotes positive feedback by linking Cdc42-GTP to its GEF Scd1." While this conclusion is consistent with the data with the Scd2 fragments, it's relevance to the full length protein is less clear. With the full length proteins, it is possible that Cdc42 recruits Scd1 via Cdc42 > Pak1 > Scd2 > Scd1, an interpretation in line with the demonstrated functionality of the Pak1-Scd1 bridge constructs. The role of the Cdc42 binding sites in Scd2 are interesting but not well established in vivo. If the authors want to claim the Cdc42 play an important role in directly recruiting Scd2, they should compare the recruitment the full length protein in the presence and absence of a point mutations that disrupt the Cdc42 binding site. This figure would be improved with a diagram of Scd2 with domains indicated.

Overall, while I appreciate the efforts the authors have made to adapt optogenetic approaches to this problem, the results they have obtained are not as compelling as they are described. Perhaps this is due to the fact that an activated allele of Cdc42 lacking its lipid tail is not an ideal probe. Cortical recruitment of many effectors of Cdc42 requires the ability to bind to both the GTPase and the membrane, certainly there are steric limitations to these interactions. Further, as the authors are acutely aware (Bendezu, 2015), tagging small GTPases is fraught with challenges. This is a case where the best approach might be to present the data along with its limitations and limit the scope of the conclusions to those that can be unambiguously drawn from the data presented.

The experiments with the Scd1-Pak bridge molecule, revealing a role for Gef1 in enhancing cell polarity and those pertaining to Ras1 are more compelling. However, a number of points related to these experiments require clarification or modification.

1 - Is the function of the "bridge" dependent upon the triple GFP tag on Scd1? This construct would be predicted to form a Scd1-(Pak1)(Pak1)(Pak1) molecule which may function differently than a Scd1-(Pak1) or, indeed, the Scd1-Scd2-Pak1 endogenous complex.

2- page 12 "Strikingly, scd2Δ gap1Δ double mutants were almost completely round and localized Scd1 and CRIB all around the membrane (Fig 7D, G)." This is not an accurate description of the images in fig 7D, particularly the second half of the sentence.

3- page 12, " These data strongly indicate that Ras1 promotes not only Scd1 localization but also its activity towards Cdc42. Thus, Ras1 acts to modulate the strength of the positive feedback." In one interpretation, this make sense, if it were a linear pathway in which Scd1 recruitment lead to CRIB recruitment. However, it is not linear, there is positive feedback, so if Ras1 only activated Scd1, then Ras1 mutants would be expected to have less Scd1 due to a reduction in positive feedback.

4- Overall the Ras1 data suggest that Ras1 primarily promotes positive feedback and has far less of an instructive role in the positioning. In the absence of Ras1 or in the absence of Gap1, the polarity complex retains the ability to localize properly. The evidence for a positional role for Ras1 far less compelling.

minor points

page 5 "Consistently" is used incorrectly. 

page 10 "Scd2K463A was "efficiently" recruited to cell sides by OptoQ61L, but was unable to induce the cell side re-localization of Scd1 (Fig 4E, S4B)". Efficiently?

Overall the authors over use adjectives: clearly, efficiently, strongly, etc. Let the data speak for themselves.

--

Reviewer #3: 

In this Manuscript the authors have developed and used the state-of-the-art optogenetic approach to investigate regulation of Cdc42 during polarization in fission yeast cells. This is the first example of such an approach being used in fission yeast. This powerful approach has the potential to significantly advance the field of cell polarization and reveal previously unidentified or poorly understood mechanism of Cdc42 activation. Indeed, this research shows for the first time in vivo evidence of a positive feedback Cdc42 activation. Moreover, the use of a Scd1-Pak1 bridge reveal some unexpected results that bring to question our previous assumptions about the mechanism of feedback pathways involved in Cdc42 regulation. While overall the study is high quality, the data clearly presented and very well suited for PLOS Biology, the authors have not considered all potential explanations for some of their observations. This leads to some strong statements without sufficient evidence to support them. The authors should consider alternate hypotheses for their observations and either resolve them or at the very least discuss them before the manuscript is accepted. Addition of a few controls may also help the manuscript as is detailed below. 

Major points

1. Pak1 kinase is involved in negative feedback regulation of Cdc42. Previous work from Lew lab and McCusker labs have shown that Pak1 phosphorylates the GEF Cdc24 to diminish GEF activity. One proposed explanation for this inhibition is that Pak1-mediated phosphorylation of Cdc24 prevents interaction with the scaffold Bem1 and in the absence of Bem1 interaction Cdc24 does not localize and cannot activate Cdc42. Here the authors made a Scd1-Pak1 bridge eliminating the need for Scd2 and this leads to localization of Scd1 and Cdc42 activation. However, with the Pak1 CRIB alone or with a kinase dead Pak1 they do not see establishment of a positive feedback or cell polarization even though Scd1 is localized to the cortex. The authors claim that this indicates that Pak1 is needed for Scd1 mediated Cdc42 activation and the positive feedback. However, this is not true since pak1 mutants (orb2-34, kinase dead and switch off) all display the same phenotype of monopolar cells with hyperactivation of Cdc42 and increased Scd1 at the single growing pole. This is likely due to lack of a negative feedback. These previous observations by multiple groups suggest that pak1 is not required for Scd1 mediated Cdc42 activation and in fact absence of Pak1 further enhances this activation. In figure 5E and F, the cells appear to show significant Scd1-3xGFP at the cortex but this is spread over a larger area. Such an observation could also be explained by the lack of Pak1 mediated phosphorylation of Scd1, where Scd1 does not accumulate at local regions and thus the cells show increased cell width. What does the CRIB signal look like in these cells? It is entirely possible that Pak1 phosphorylates Scd1 only when they are in a complex and this phosphorylation tunes Cdc42 activation. In such a scenario the wild type Pak1 present in these cells will not associate with Scd1 (especially since Scd2 is absent). 

2. Another unexpected and fairly interesting result is that the Gef1-Pak1 bridge does not act the same way as the Scd1-Pak1 bridge. The authors claim that this indicates that Scd1 is the necessary for positive feedback. In figure 7 F the authors show Gef1-Pak1 bridge does not restore polarity in scd1Δgap1Δ mutants. It is not clear how Gef1 localizes to the cortex in scd2Δgap1Δ mutants without the bridge. This control would explain if interaction with Pak1 impairs Gef1 activity. Could it be that Gef1 localization or activity is impaired upon forced Pak1 interaction? Would a Gef1 bridged with the Pak1 CRIB domain also show the same phenotype. Would a constitutively localized gef1S112A mutation (Das et al 2015) when bridged to Pak1 also show the same phenotype? 

Minor points

1. Throughout the paper the authors use the term GEF when they specifically mean Scd1. This can be a bit confusing to the readers and should be avoided as much as possible. 

2. In the abstract the authors claim that “Ras1 GTPase plays a dual role in localizing and activating the GEF….”. This is a strong statement as we do not know how Ras1 promotes GEF activity, it could function by activating the GEF or facilitating GTPase GEF interaction or by some other unknown means. A more precise wording would be “Ras1 GTPase plays a dual role in localizing the GEF and promoting its activity….”. 

3. In page 2 last line, Cdc42 localization was also reported by Murray and Johnson 2000, and Coll et al, 2007. These references should be cited. 

4. In page 29 the authors state that “sum projections of five consecutive images are shown”. Do they mean 5 frames along the Z axis? This should be clarified. 

5. In figure 2 the control shown in panel B is not explained wither in the legend or in the text but only explained in the methods. A single sentence explaining the control either in the results or in the legend will help the reader. 

6. Figure 3A, the black arrowheads are not very clear in a grayscale image. Can the authors change the arrowheads to a more obvious color?

--

Reviewer #4: 

In this study, Lamas et. al. study how the spatial localization and activity of the master polarity regulator Cdc42 is controlled in fission yeast. In their study, they modify a number of optogenetic tools for use in fission yeast, an important ‘first’ for this model system. Past studies have shown that Cdc42 activity is controlled by activating factors such as the GEFs Gef1 and Scd1, and also by inactivating factors such as a variety of GAP proteins. In cylindrical fission yeast cells, spatial localization of these proteins helps ensure that active Cdc42 is restricted to the cell ends, which are the sites of polarized growth in fission yeast. If active Cdc42 is mis-localized to the cell sides, cells grow to the wrong shape due to growth in the wrong places. A number of past studies in budding yeast have led to a feedback model, whereby active Cdc42 promotes the recruitment of its activators, namely a scaffold (Bem1, called Scd2 in fission yeast) and a GEF (Cdc24, called Scd1 in fission yeast). Lamas et al demonstrate that Scd1 can be recruited to Cdc42 through two distinct adaptor proteins: Scd2 or the GTPase Ras1. Next, using an optogenetic strategy that leverages the photo-sensitive Cry2-CIB interaction, these authors show that ectopically targeting a constitutively active Cdc42 mutant to the plasma membrane results in recruitment of Cdc42 activators and effectors, which is consistent with a positive feedback mechanism through which Cdc42 promotes its own activation. Finally, the authors demonstrate that the primary function of the scaffold protein Scd2 is to lead to recruitment of the essential Cdc42 effector Pak1 kinase.

This study has a number of clear strengths. This study represents the first use of any optogenetic system in fission yeast, at least to my knowledge. The authors do an excellent job of characterizing and validating the robustness of the Cry2-CIB optogenetic system in fission yeast, which will be a strong and broadly applicable tool for the fission yeast community. They also mention in the discussion that other systems did not work in their hands, which could be quite helpful to other labs interested in this approach. While optogenetics has been employed in budding yeast to study the regulation of Cdc42, the new tools developed in this manuscript will enable future studies of Cdc42 in fission yeast, which is comparatively understudied. Additionally, the technical experimental quality of this work is very high, both in the characterization of the optogenetic system and its application to the study of Cdc42 regulation. The experimental design is elegant and clever, and will be of use for studying many related questions.

Some aspects of the study might limit its overall impact. Primarily, the notion of positive feedback in this system has been studied extensively in budding yeast, and the authors’ conclusions largely fit within the preestablished model. It is worth noting that they are the first to provide strong evidence for this feedback in fission yeast, so the work is likely to be cited widely as evidence for conservation of this feedback, but the conceptual novelty is limited. I also found their experiments and results are largely consistent with positive feedback, but perhaps not as positively conclusive as they have sold it in the text. From my reading of the paper, I felt that the coordination of Ras1 and Scd2 activities represented the most novel finding, but it was not explored in the same depth as the positive feedback model.

Major comments/concerns

1. In some parts of the manuscript, the conclusions are seemingly at odds with each other. For example, the authors state that “Scd1 recruitment to activated Cdc42 fully depends on Scd2”, but the data suggest that Ras1 also functions in recruitment of Scd1 to Cdc42. Finally, the most novel result in this manuscript is that “Ras1 cooperates with Scd2 for Scd1 recruitment to cell poles”.

2. The authors refer to “dynamic patches formed at the membrane (Figure 1F, Movie 1)” of Cdc42. These dynamics should be quantified and not just shown in a supplemental movie. Parameters such as zone formation frequency and duration would be of interest.

3. The abstract and introduction should be reworked for clarity and logical flow. For the non-aficionado of Cdc42, the introduction is very hard to follow. The discussion of feedback and past studies in this section makes it even more dense and confusing. There are also a number of typos.

4. On Page 8, the authors state that “Because active Cdc42 recruits is own activator Scd1 along with the scaffold protein Scd2 and the effector protein Pak1, this directly establishes the existence of a positive feedback…”. I would not argue that the conclusion has been so strongly proven. For example, the authors have not tested whether WT, or catalytically inactive Cdc42 can also recruit these factors. Therefore, these results could be explained by a mechanism not requiring positive feedback. One option for improvement would be to perform these experiments with the Opto-WT Cdc42 strain for comparison. Given that positive feedback is a central theme of this manuscript, it is important that this result be firmly established in the data. If WT or a GDP-locked Cdc42 recruit Scd1/Pak1 as efficiently as the GTP locked form, then there is no positive feedback and the result stems from simpler protein-protein interaction mechanisms. In this vein, the experiments in Figure S5B suggest that not all Cdc42 regulators and effectors are recruited in a manner consistent with positive feedback. The authors are well positioned to determine which proteins (Pak1, Gef1,Scd1, Scd2,Ras1) are recruited by positive feedback and which are not using their optogenetic tools.

Minor comments/concerns

1. Some form of data behind Table S1 should be provided.

2. In Figure 2G, optogenetic recruitment of GTP locked Cdc42 but not WT leads to cell rounding. Why is the WT seemingly completely inactive?

3. Why does expression of CRY2PHR-Cdc42Q61L result in nuclear localization of Cdc42 and its regulators to the nucleus? Does this impact any of the experiments?

4. The authors state that “Scd1 recruitment to activated Cdc42 fully depends on Scd2”, but the data suggest that Ras1 also functions in recruitment of Scd1 to Cdc42. This confusion should be resolved.

5. On Page 10, the authors write that “…the main function of Scd2 is to mediate GEF-PAK complex formation”. This is confusing given the result that Scd2 is not required for Pak1 recruitment to zones of active Cdc42. The authors should clarify their hypothesis here.

6. The conclusion on P.10 that “lethality suppression was specifically due to feedback restoration and not simply to Scd1 re-localization to cell ends” seems overly strong. As stated above, the results are consistent with positive feedback as well as other mechanisms. Also, the use of Tea1 and For3 are strange controls here (at least for me) because these are different complexes that would not be expected to combine Scd1 and Pak1 in the same complex. Either the data should be strengthened, or the language softened.

7. In the conclusions (P.17 and Fig 7J) the authors make a number of assertions that Pak1 localization is critical for relaying positional information to Cdc42, and operates in parallel with Cdc42 positive feedback. These assumptions seem highly testable and relevant to this work.

---

## [Decision Letter · Decision Letter 2]

2 Dec 2019

Dear Sophie,

Thank you for submitting your revised Research Article entitled "Optogenetics reveals Cdc42 local activation by scaffold-mediated positive feedback and Ras GTPase" for publication in PLOS Biology. I have now obtained advice from the original reviewers and have discussed their comments with the Academic Editor. As you can see, the reviewers are happy with the revision and we're delighted to let you know that we're now editorially satisfied with your manuscript. 

However before we can formally accept your paper and consider it "in press", we also need to ensure that your article conforms to our guidelines. A member of our team will be in touch shortly with a set of requests. As we can't proceed until these requirements are met, your swift response will help prevent delays to publication. Please also make sure to address the data and other policy-related requests noted at the end of this email.

*Copyediting*

*Published Peer Review History*

*Early Version*

*Submitting Your Revision*

Sincerely,

Hashi Wijayatilake, PhD, 

Managing Editor

PLOS Biology

DATA POLICY:

Figs. 1BDEG, 2DEF, 3CDEF, 4ABD, 5BDE, 6CDEF, 7BCDF, 8F, 9BD, S2B, S4AB, S5A-D, S6AB, S7AB, S8B, S9B

REVIEWS:

Reviewer #1: 

This already strong paper has been further enhanced by the revisions. The authors have satisfactorily addressed all questions. This is an excellent paper and represents an important advance.

Reviewer #2: 

The authors have done a comprehensive job responding to the reviewers comments. I now support publication. I am not fully convinced that the optogenetically recruited Cdc42 is inducing physiological positive feedback - in part because so much Cdc42 is recruited to the membrane due to the global activation, but the data shown does support the conclusions drawn.

Reviewer #3: 

The authors have successfully addressed all the issues raised by me and the additional data strengthen their story. This manuscript is suitable for PLOS BIOLOGY.

Reviewer #4: 

The authors have addressed most of the reviewer concerns, and have put together a nice paper that will be useful to the field.

---

## [Editor Report · Decision Letter 3]

2 Jan 2020

Dear Dr Martin,

On behalf of my colleagues and the Academic Editor, Nicolas Tapon, I am pleased to inform you that we will be delighted to publish your Research Article in PLOS Biology. 

Early Version

PRESS 

Kind regards,

Hannah Harwood

Publication Assistant, 

PLOS Biology

on behalf of

Hashi Wijayatilake,

Managing Editor

PLOS Biology